# N-6-methyladenosine (m6A) promotes the nuclear retention of mRNAs with intact 5′ splice site motifs

Eliza S Lee[1], Harrison W Smith[1], Yifan E Wang[1], Sean SJ Ihn[1], Leticia Scalize de Oliveira[1], Nevraj S Kejiou[1], Yijing L Liang[2], Syed Nabeel-Shah[3,4], Robert Y Jomphe[1,5] (iD), Shuye Pu[4], Jack F Greenblatt[3,4] (iD), Alexander F Palazzo[1] (iD)

In humans, misprocessed mRNAs containing intact 5′ Splice Site (5′SS) motifs are nuclear retained and targeted for decay by ZFC3H1, a component of the Poly(A) Exosome Targeting complex, and U1-70K, a component of the U1 snRNP. In *S. pombe*, the ZFC3H1 homolog, Red1, binds to the YTH domain–containing protein Mmi1 and targets certain RNA transcripts to nuclear foci for nuclear retention and decay. Here we show that YTHDC1 and YTHDC2, two YTH domain-containing proteins that bind to *N*-6-methyladenosine (m6A) modified RNAs, interact with ZFC3H1 and U1-70K, and are required for the nuclear retention of mRNAs with intact 5′SS motifs. Disruption of m6A deposition inhibits both the nuclear retention of these transcripts and their accumulation in YTHDC1-enriched foci that are adjacent to nuclear speckles. Endogenous RNAs with intact 5′SS motifs, such as intronic polyadenylated transcripts, tend to be m6A-modified at low levels. Thus, the m6A modification acts on a conserved quality control mechanism that targets misprocessed mRNAs for nuclear retention and decay.

## Introduction

Eukaryotic cells are divided into two compartments: the nucleus where pre-mRNAs are synthesized and processed and the cytoplasm where mature mRNAs are translated into proteins. This subcellular division allows for extensive quality control at the level of both RNA synthesis and RNA processing to promote the nuclear retention and degradation of spurious transcripts and misprocessed mRNAs before they access the translation machinery in the cytoplasm (Palazzo & Lee, 2018; Garland & Jensen, 2020; Palazzo et al, 2024). Misprocessed mRNAs typically contain intact RNA processing *cis*-elements which promote retention and decay (Legrain & Rosbash, 1989; Abad et al, 2008; Lee et al, 2015).

One class of misprocessed mRNAs are intronic polyadenylated (IPA) transcripts (Tian et al, 2007). These are generated by cryptic 3′cleavage/polyadenylation signals found in introns. Although most of these cryptic 3′cleavage/polyadenylation signals are suppressed (Kaida et al, 2010), they can be used at low levels to generate IPA transcripts that tend to encode protein fragments (Ogami et al, 2017), which are often toxic as they can compete with endogenous proteins for their binding partners (Veitia, 2007). IPA transcripts contain intact 5′ splice site (5′SS) motifs, which are present at the boundary between the upstream exon and the intron where the 3′ cleavage occurs. We and others have shown that the presence of an intact 5′SS motif in an RNA promotes its nuclear retention and decay (Legrain & Rosbash, 1989; Abad et al, 2008; Lee et al, 2015). These transcripts are recognized by the U1 snRNP, and the U1 component U1-70K is required for their nuclear retention and decay (Fortes et al, 2003; Abad et al, 2008; Goraczniak et al, 2009; Lee et al, 2022). We previously found that U1-70K associates with ZFC3H1 (Lee et al, 2022), a component of the Poly(A) Exosome Targeting (PAXT) complex. Moreover, we and others have shown that ZFC3H1 is required for the nuclear retention and decay of RNAs with intact 5′SS motifs including IPA transcripts (Ogami et al, 2017; Lee et al, 2022). PAXT also contains the nuclear poly(A) binding protein, PABPN1, and the RNA helicase MTR4, which is known to feed RNAs into the nuclear exosome, the major RNase complex in human cells (Meola et al, 2016; Ogami et al, 2017; Ogami & Manley, 2017; Silla et al, 2018, 2020). Although MTR4 is not required for nuclear retention, PABPN1 likely plays a role (Lee et al, 2022).

The PAXT complex is conserved between humans and *S. pombe*, where it is known as the MTREC (Mtl1-Red1 Core) complex, and is composed of Mtl1, the paralogue of Mtr4, Red1, the paralogue of ZFC3H1 and Pab2, a poly(A) binding protein (Sugiyama & Sugioka-Sugiyama, 2011; Zhou et al, 2015; Shichino et al, 2020). The MTREC complex promotes the nuclear retention and decay of meiotic mRNAs that are inappropriately synthesized in interphase cells. MTREC is recruited to these RNAs through a complex of three proteins consisting of Erh1, Pir1, and Mmi1 (Sugiyama & Sugioka-

[1]Department of Biochemistry, University of Toronto, Toronto, Canada   [2]Centre for Computational Medicine, Hospital for Sick Children, Toronto, Canada   [3]Department of Molecular Genetics, University of Toronto, Toronto, Canada   [4]Terrence Donnelly Centre for Cellular and Biomolecular Research, Toronto, Canada   [5]Cell Biology Program, Hospital for Sick Children, Toronto, Canada

Correspondence: alex.palazzo@utoronto.ca
Eliza S Lee's present address is Department of Biochemistry, University of Colorado Boulder, Boulder, CO, USA

Sugiyama, 2011; Sugiyama et al, 2016; Shichino et al, 2020; Wei et al, 2021). Note that Pir1 has a Serine/Proline-rich region that shows a limited homology with the N-terminus of ZFC3H1 (Wei et al, 2021) and that Mmi1 contains a YTH domain, which in other systems recognizes *N*-6-methyladenosine (m6A) modified RNAs. In *S. pombe*, no m6A modification is detectable in mRNAs (Ishigami et al, 2021) and instead Mmi1 binds to specific RNA motifs (Chen et al, 2011; Sugiyama & Sugioka-Sugiyama, 2011; Wang et al, 2016). MTREC and Mmi1 are also involved in the nuclear retention and decay of non-coding RNAs and transposable element-derived RNAs (Zhou et al, 2015; Sugiyama et al, 2016; Vo et al, 2019). When RNA decay is inhibited, MTREC-targeted RNAs accumulate in Mmi1/Red1-enriched nuclear foci (Sugiyama & Sugioka-Sugiyama, 2011; Shichino et al, 2018, 2020).

In humans, YTH domains recognize m6A-modified RNAs (Xu et al, 2014). This modification is enriched in several regions of mRNAs, including large exons, regions surrounding the stop codon, and 3′UTRs (Dominissini et al, 2012; Meyer et al, 2012), and is depleted around exon-exon junctions (Yang et al, 2022; He et al, 2023; Luo et al, 2023; Uzonyi et al, 2023). The m6A modification is catalyzed by the METTL3 methyltransferase, which forms a complex with METTL14 and other factors (Liu et al, 2014). This modification can be removed by the demethylases ALKBH5, FTO, and perhaps ALKBH3 (Jia et al, 2011; Zheng et al, 2013; Ueda et al, 2017). m6A is predominantly recognized by five YTH domain–containing proteins in humans, including YTHDC1 and YTHDC2, which are localized to the nucleoplasm (Xu et al, 2021).

Initial studies indicated that the m6A modification, and m6A-binding proteins promote mRNA export (Zheng et al, 2013; Roundtree et al, 2017; Lesbirel et al, 2018; Lesbirel & Wilson, 2019); however, more recent work has found that the levels of m6A modification correlate with nuclear retention (Tang et al, 2024). Moreover, m6A and YTHDC1 are required for the suppression of RNAs generated from transposable elements, enhancer RNAs and other non-coding RNAs (Liu et al, 2020), reminiscent of the function of MTREC, and Mmi1 in *S. pombe*. Like Mmi1 and Red1, YTHDC1 and ZFC3H1 are found in nuclear foci that contain mRNA (Nayler et al, 2000; Cheng et al, 2021; Wang et al, 2021). These foci partially overlap nuclear speckles, which are required for the nuclear retention of mRNA with intact 5′SS motifs (Lee et al, 2022).

Here, we show that m6A modification is required for the efficient nuclear retention of mRNAs that contain intact 5′SS motifs, which are found in IPA transcripts, and retains these in nuclear foci present adjacent to nuclear speckles.

## Results

### YTHDC1 and YTHDC2 interact with both ZFC3H1 and U1-70K

In *S. pombe*, the MTREC complex associates with Mmi1, a nuclear YTH domain-containing protein (Fig 1A, left) (Chen et al, 2011; Sugiyama & Sugioka-Sugiyama, 2011; Zhou et al, 2015; Shichino et al, 2020). To determine whether components of the equivalent human complex (known as PAXT, Fig 1A right) interact with nuclear YTH domain-containing proteins, we expressed either FLAG-HA-YTHDC1

or FLAG-ZFC3H1-HA in HEK293 cells and assessed their binding partners by co-immunoprecipitation in the absence or presence of RNase, which removed all detectable RNA from the lysate (Fig S1A). In FLAG-HA-YTHDC1 immunoprecipitates, we detected both endogenous ZFC3H1 and U1-70K, even in the presence of RNase (Figs 1B and S1B). FLAG-HA-YTHDC1 immunoprecipitates also contained YTHDC2 but not any detectable ALKBH5 (Figs 1B and S1B). In FLAG-ZFC3H1-HA immunoprecipitates, we detected endogenous YTHDC1, YTHDC2, and U1-70K, even in the presence of RNase, but not eIF4AIII or G3BP2 (Figs 1C and D and S1C). In all cases, the plasmid-expressed proteins were present at levels that were less than 10-fold the endogenous protein level and had only minor effects on the levels of their interacting partners (Fig S1D–I). When we immunoprecipitated a smaller fragment of FLAG-ZFC3H1 (FLAG-ZFC3H1 1-1233), which lacks the C-terminal TPR repeats but contains all known protein interaction-domains (Wang et al, 2021), we again detected YTHDC1, YTHDC2, and U1-70K in the precipitates, but not ALKBH5, eIF4AIII, METTL3, or ALY (Fig 1E and F). These results indicate that nuclear YTH proteins associate with ZFC3H1 and U1-70K, two factors required for the nuclear retention of RNAs that contain intact 5′SS motifs.

### DRACH motifs are required for the efficient nuclear retention of mRNAs with an intact 5′SS motif

The m6A methyltransferase METTL3 preferentially modifies RNAs at the consensus DRACH motif (G/A/U, A/G, A, C, A/U/C) (Dominissini et al, 2012; Meyer et al, 2012). To test whether m6A was required for nuclear retention, we mutated the DRACH motifs in the *fushi-tarazu* (*ftz*) reporter which we previously used to study the effect of the 5′SS motif on nuclear retention (Lee et al, 2015). For the parental intronless *ftz* reporter (*ftz-Δi*), we identified 10 putative DRACH motifs, and for the 5′SS motif containing version (*ftz-Δi-5′SS*), we found 11 motifs (Fig 2A). We mutated all but of one of the DRACH sites to generate two other reporters, *no-m6A-ftz-Δi* and *no-m6A-ftz-Δi-5′SS*, respectively. Note that one DRACH motif, present near the consensus 5′SS motif, was not altered. When the *ftz-Δi-5′SS* reporter was expressed in human osteosarcoma cell lines (U2OS), elimination of most DRACH motifs drastically reduced m6A deposition (Fig 2B). When the distribution of the mRNAs produced from these various reporters was assessed by RNA FISH, we found that although the presence of an intact 5′SS motif promoted nuclear retention, elimination of most DRACH motifs inhibited this retention (Fig 2C and D). Elimination of the DRACH motifs in *ftz-Δi* had no effect on its export (Fig 2C and D). When we expressed versions of *ftz-Δi* and *ftz-Δi-5′SS* mRNA with optimized DRACH motifs (AGACT) to enhance methylation (creating *e-m6A-ftz-Δi* and *e-m6A-ftz-Δi-5′SS*; Fig 2A), we did not observe a significant change in their distribution (Fig 2D).

### The m6A methyltransferase, METTL3, is required for the nuclear retention of mRNAs with an intact 5′SS motif

Next, we tested whether the small molecule STM2457, which sterically blocks the S-adenosylmethionine-binding site of METTL3 and inhibits its RNA methylation activity (Yankova et al, 2021), also inhibited nuclear retention of mRNA with intact 5′SS motifs. When

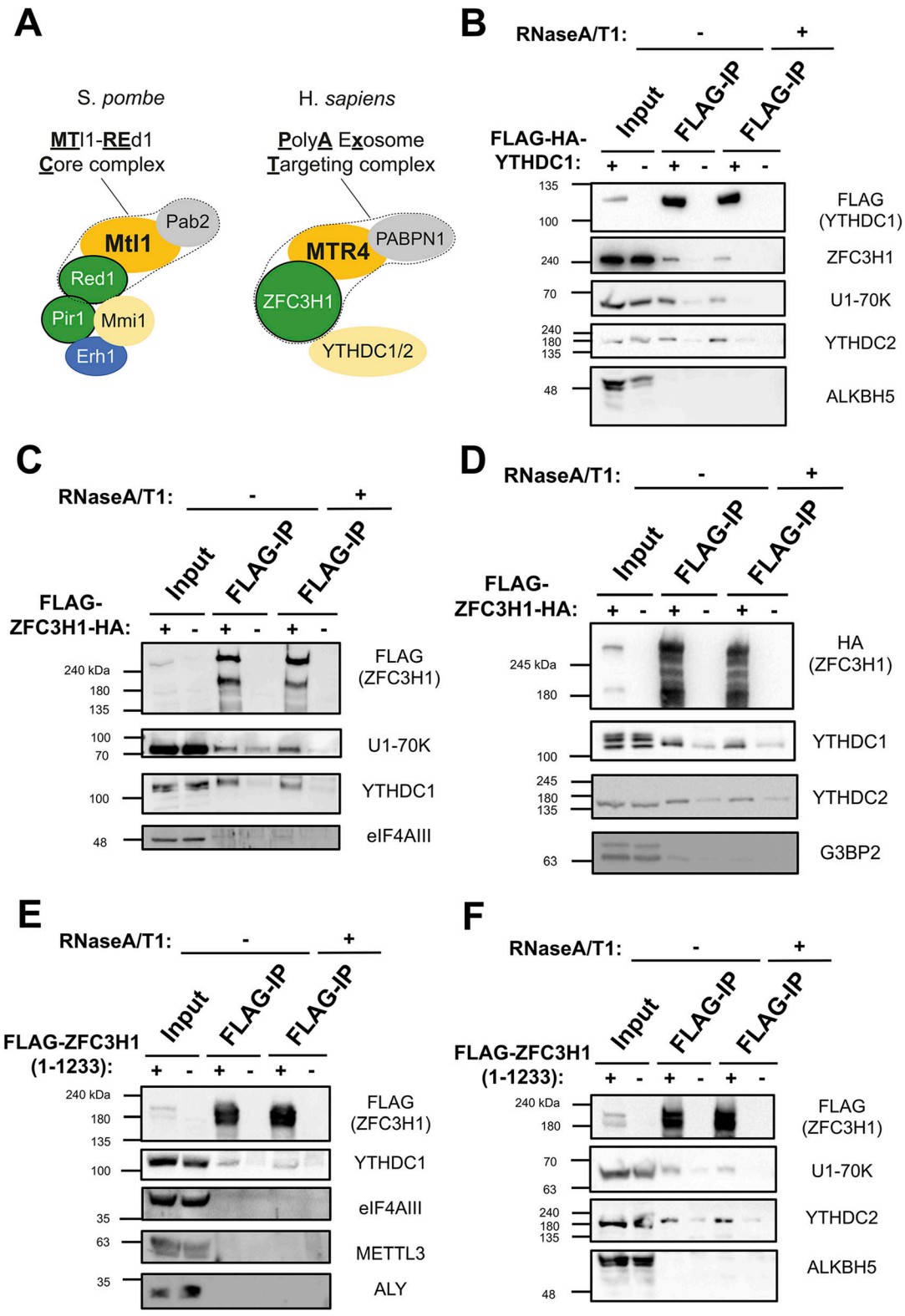

**Figure 1. YTHDC1 and YTHDC2 interact with ZFC3H1 and U1-70K.**
**(A)** The main components of the MTREC complex (outlined by a dotted line) and associated factors in *S. pombe* (left), and their human homologs, including the PAXT complex (outlined by a dotted line) and YTH proteins (right). **(B)** HEK293 cells were either transfected with FLAG-tagged YTHDC1 or mock transfected. After 18–24 h, cell lysates were collected and subjected to immunoprecipitation using FLAG M2 beads. Precipitates were then treated with either buffer or a combination of RNase A and RNase T1 and reisolated to remove RNA-dependent interacting proteins. Samples were separated by SDS–PAGE and immunoblotted for the indicated proteins. Note

U2OS cells were treated with 100 μM STM2457 for 18 h, *ftz-Δi-5'SS* mRNA was no longer nuclear retained when compared with DMSO-treated cells (Fig 2E and F). Note that STM2457-treatment had no effect on the nuclear export of *ftz-Δi* mRNA (Fig 2E and F) and minimal effects on the levels of ZFC3H1, YTHDC1, YTHDC2, or U1-70K (Fig S2A and B).

To further confirm this result, we depleted METTL3 using a combination of four lentiviral-delivered shRNAs (Fig 2G). We found that in METTL3-depleted cells, *ftz-Δi-5'SS* mRNA was more cytoplasmic than in cells treated with control shRNA (Fig 2H and I). Again, the nuclear export of *ftz-Δi* mRNA was not affected. In aggregate, these results demonstrate that the m6A modification is required for the nuclear retention of reporter mRNAs containing intact 5'SS motifs.

### YTHDC1 and YTHDC2 are required for the nuclear retention of 5'SS motif containing reporter mRNAs

To test whether YTHDC1 and YTHDC2 are required for the nuclear retention of mRNAs with intact 5'SS motifs, we co-depleted these two proteins (Fig 3A) using two lentivirus-delivered shRNAs for each gene. In parallel, we depleted ZFC3H1, which is required for the retention of mRNAs with intact 5'SS motifs (Lee et al, 2022), and ALKBH5, an m6A demethylase (results described in the next section; Fig 3A).

We observed that in YTHDC1/2-co-depleted cells, *ftz-Δi-5'SS* mRNA was no longer efficiently retained in the nucleus when compared with control shRNA-treated cells (Fig 3B and C). The degree to which nuclear retention was inhibited was similar to ZFC3H1-depleted cells (Fig 3B and C). Both YTHDC1/2-co-depletion and ZFC3H1 depletion had no effect on the nuclear export of our reporter mRNAs that lacked either the 5'SS motif (*ftz-Δi*) or most DRACH motifs (*no-m6A-ftz-Δi-5'SS*; Fig 3C). Note that YTHDC1/2-depletion had some effect on the levels of ZFC3H1, and vice versa (Fig S3A and B).

Next, to validate our results with more natural IPA-generating genes, we monitored IPA transcripts generated from two mini genes, *PCF11-IPA* and *CCDC71-IPA* (Figs 3D–F and S3C–E), that we had previously characterized (Lee et al, 2022). Both reporters are based on endogenous IPA transcripts and require both ZFC3H1 and U1-70K for their efficient nuclear retention. These IPA transcripts can then be specifically monitored by FISH using fluorescent probes that hybridize into the intronic region, just upstream of the 3' cleavage/polyadenylation site (Figs 3D and S3C). In agreement with our previous results, the two reporters were more cytoplasmic in cells co-depleted of YTHDC1 and YTHDC2 when compared with cells treated with control shRNA (Figs 3E and F and S3D and E).

To examine the effects on endogenous IPA transcripts, we fractionated U2OS cells (Fig S3F) that were treated with various shRNAs and assessed the levels of endogenously generated *PCF11-IPA* transcripts by qRT-PCR. Depletion of either ZFC3H1 or YTHDC1/2 caused an increase in the total levels and in the cytoplasmic/

nuclear ratios of endogenous *PCF11*-IPA transcripts (Fig 3G and H), although depletion of the former had a greater effect than depletion of the latter. Collectively, these results suggest that YTHDC1 and YTHDC2 are required for the efficient nuclear retention of mRNAs with intact 5'SS motifs.

### Depletion of the m6A demethylase ALKBH5 inhibits nuclear retention of mRNAs with intact 5'SS motifs

We reasoned that depletion of enzymes that remove the m6A modification should enhance the nuclear retention of transcripts with intact 5'SS motifs. To test this, we depleted ALKBH5 in U2OS cells using lentiviral-delivered shRNAs (Fig 3A). Note that because of the nature of our surprising results (see below), we used two different shRNAs to ensure that our findings were not because of off-target effects.

Unexpectedly, we found that depletion of ALKBH5 inhibited the nuclear retention of *ftz-Δi-5'SS* mRNA (Fig 3B and C), *PCF11-IPA* (Fig 3E and F), and to a lesser extent *CCDC71-IPA* (Fig S3D and E). Depletion of ALKBH5 had no effect on the nuclear export of our reporter mRNAs that lacked either the 5'SS motif (*ftz-Δi*) or DRACH motifs (*no-m6A-ftz-Δi-5'SS*) (Fig 3B and C). Consistent with previous reports (Zheng et al, 2013), we found that ALKBH5 was present in the nuclear fraction, although it was also present in the cytoplasm (Fig S3G).

From these results, we concluded that ALKBH5 was required for the efficient nuclear retention of most mRNAs that contained intact 5'SS motifs. Although this result seems counter to our other findings, it may indicate that ALKBH5 may play additional roles beyond removing m6A modifications (see the Discussion section).

### m6A accumulates in IPA transcripts

We next asked whether m6A methylation marks accumulate within natural IPA transcripts. We analysed CLIP-seq experiments which used an m6A-specific antibody to enrich modified RNAs isolated from HEK293 cells. We compared the distribution of these m6A methylation marks in intronic regions that have been previously shown to generate IPA transcripts which are sensitive to ZFC3H1 depletion (Lee et al, 2022). As a comparison, each intronic region in an IPA transcript was randomly matched with an exon that was both length- and expression-matched as determined by high through-put sequencing. This matching process was repeated five times across two separate transcriptome sequencing experiments. Although we observed that the average number of m6A modifications per RNA was similar between the intronic regions of IPA transcripts ("IPA" in the graph) and the length/expression-matched exons (Fig 4A), significantly fewer of the former lacked detectable m6A modifications (Fig 4B) and significantly more had a low number of detectable m6A modifications (1–6 sites) when compared with length/expression-matched exons. In contrast, significantly fewer

---

that the inputs represent 5% of the samples used for immunoprecipitation. **(C, D)** Similar to (B), except that cells were transfected with FLAG/HA-tagged ZFC3H1 (FLAG-ZFC3H1-HA). **(E, F)** Similar to (B), except that cells were transfected with a FLAG-tagged fragment of ZFC3H1 containing amino acids 1–1,233.
Source data are available for this figure.

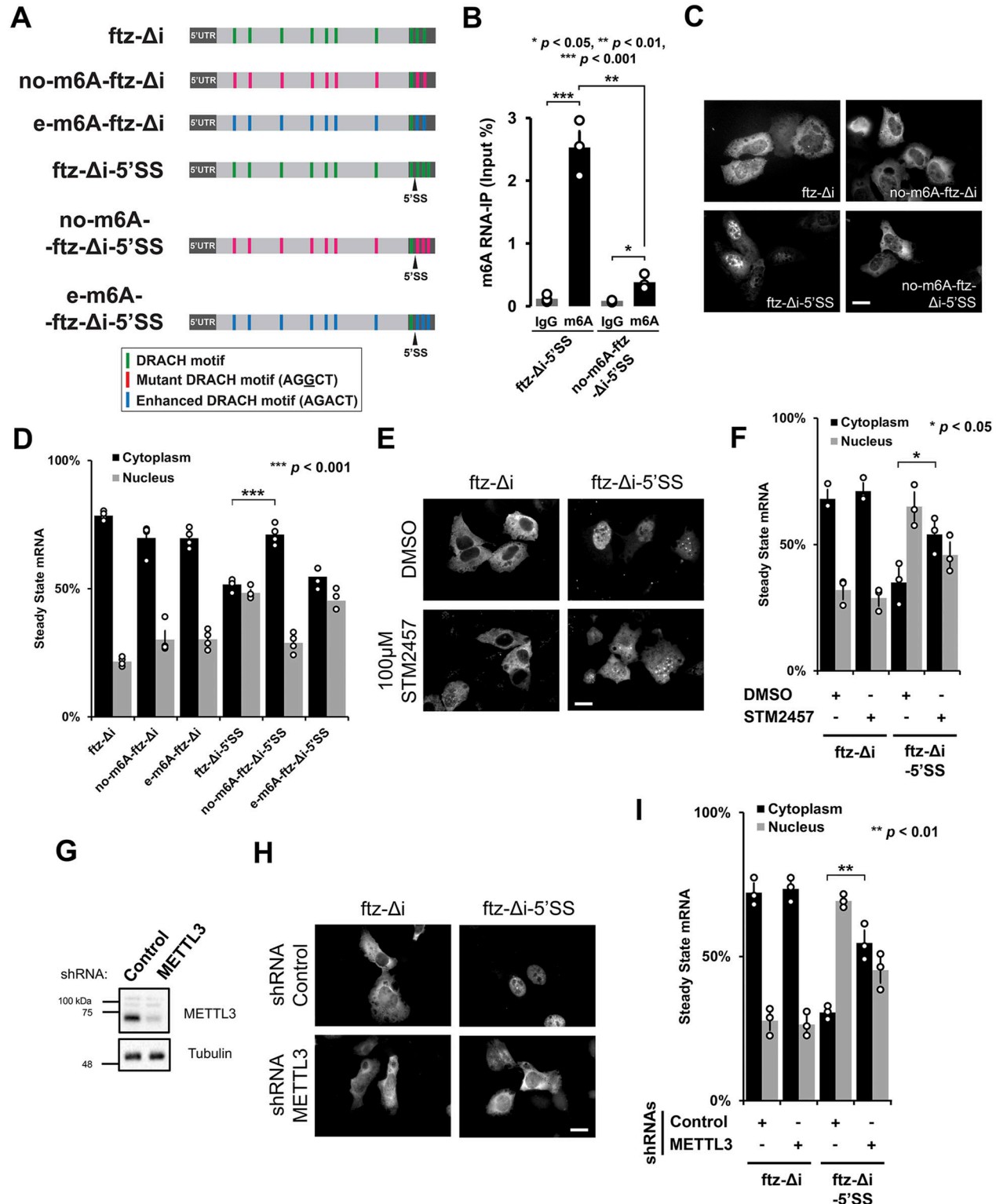

**Figure 2.  The m6A modification is required for nuclear retention of 5'SS motif containing mRNAs.**
**(A)** Schematic of the intronless fushi-tarazu (*ftz-Δi*) reporters used in this study, with and without the 5'SS motif in the 3'UTR. Note that the consensus 5'SS motif promotes nuclear retention. The positions of either consensus DRACH motifs, or their mutated forms that either eliminate the motifs (AGGCT), or enhance the motifs (AGACT) are indicated. **(B)** U2OS cells were transfected with the indicated *ftz* reporters. After 18–24 h, RNA was isolated and immunoprecipitated with anti-m6A antibodies or IgG controls. Levels of *ftz* in the precipitates were assessed by qRT-PCR. Each bar represents the average and standard error of three independent experiments. *t* test was performed, *P < 0.05 **P < 0.01 ***P < 0.001. **(C, D)** U2OS cells were transfected with the indicated *ftz* reporters. 18–24 h later the cells were fixed, and the mRNA was visualized by FISH using probes against *ftz*. Representative images are shown in (C), scale bar = 20 μm, and quantification is shown in (D). Each bar represents the average

intronic regions in IPA transcripts had a high number of m6A modifications (>12 sites) when compared with length/expression-matched exons (Fig 4B).

When the density of m6A modifications was averaged along the length of the intronic portion of IPA transcripts, we found that these m6A marks were slightly enriched near the start of the intronic portion of the IPA (i.e., near the 5′SS motif) and near the 3′cleavage/polyadenylation site, when compared with the density of m6A averaged along all exons, where m6A was depleted at both ends near exon-exon boundaries (Fig 4C) as seen by other groups (Yang et al, 2022; He et al, 2023; Luo et al, 2023; Uzonyi et al, 2023). When we looked more closely at three example genes that generate IPA transcripts, *PCF11*, *CSNK1E*, and *NCOA5*, we found that m6A marks were present throughout the intronic region that generates the IPA transcript (Fig 4D–F). In some cases, as in the *PCF11* and *NCOA5* IPA transcripts, there were only a few m6A marks, but these tended to be near one of the ends of the IPA transcript and in the upstream exon. Note that the level of all these IPA transcripts increases in ZFC3H1-depleted U2OS cells (Fig 4D–F) as previously reported (Lee et al, 2022).

### mRNAs with intact 5′SS motifs accumulate in nuclear foci containing YTHDC1, and this requires DRACH motifs

Previously, we found that 5′SS motif containing mRNAs are targeted and then retained in nuclear foci and that their nuclear retention requires nuclear speckles (Lee et al, 2015, 2022). We also found that ZFC3H1 and U1-70K are dispensable for the targeting of these mRNAs to nuclear speckles but instead prevent these RNAs from exiting nuclear foci (Lee et al, 2022). With this in mind, we asked if m6A modifications affect the trafficking of mRNAs in and out of nuclear foci.

To determine the localization of mRNAs at early time points post-synthesis, plasmids containing *ftz-Δi-5′SS* and *no-m6A-ftz-Δi-5′SS* reporter genes were microinjected into the nuclei of U2OS cells to allow for rapid synchronous expression. After allowing for 1 h of mRNA expression, a time point when nuclear speckle targeting is highest (Akef et al, 2013), cells were fixed and immunostained for SC35, a nuclear speckle marker (Spector et al, 1991) and probed for *ftz* mRNA by FISH. We observed robust nuclear speckle targeting of both constructs (Fig 5A and B). This result indicated that within the context of *ftz-Δi-5′SS*, m6A was not required for nuclear speckle targeting.

Next, we assessed whether m6A affected the trafficking of mRNAs with intact 5′SS motifs out of nuclear speckles. We thus monitored the mRNA distribution 24 h after transfection of the plasmids with reporter genes. At this time point, the distribution of the reporter mRNA reaches steady state and localization in foci is dictated by the rates of entry and egress (Akef et al, 2013). In these cells, *ftz-Δi-5′SS* was mostly localized in foci that contained YTHDC1 (Fig 5C, see arrows in the first row). These foci were adjacent to nuclear speckles. When *no-m6A-ftz-Δi-5′SS* mRNA was visualized, we observed a slight but consistent colocalization with the nuclear speckle marker SC35, despite the fact that this mRNA was well exported (Fig 5C). There was a lower level of colocalization between this mRNA and YTHDC1. These trends were confirmed by Pearson correlation analysis (Fig 5D). These results indicate that m6A promotes the transfer of mRNAs from nuclear speckles to nearby YTHDC1-containing nuclear foci (Fig 5E).

## Discussion

Here we show that m6A modifications and the m6A-binding proteins, YTHDC1 and YTHDC2, are required for the nuclear retention of mRNAs containing intact 5′SS motifs. Moreover, we find that the m6A-binding proteins YTHDC1 and YTHDC2 associate with ZFC3H1 and U1-70K. It is thus likely that these factors form a complex that targets these misprocessed mRNAs for nuclear retention and possibly degradation by the nuclear exosome, which is known to receive substrate RNAs from the PAXT complex. Recent work has demonstrated that m6A modifications are excluded from regions surrounding exon-exon splice site junctions by the exon–junction complex (Yang et al, 2022; He et al, 2023; Luo et al, 2023; Uzonyi et al, 2023). Thus, m6A modifications accumulate on unprocessed or misprocessed RNAs, and in conjunction with intact 5′SS motifs, likely recruit ZFC3H1 to promote RNA nuclear retention and decay (Fig 5E). Furthermore, our findings suggest that mRNAs with 5′SS are first targeted to nuclear speckles and then transferred to YTHDC1-enriched foci, a nuclear structure that has been likely documented by other groups (Silla et al, 2018; Cheng et al, 2021; Wang et al, 2021).

Our new results suggest that m6A modifications act as a layer of quality control to suppress the expression of unwanted transcripts. Previous reports have implicated m6A in promoting mRNA export, and this is partially based on the analysis of ALKBH5-depletion, which we believe has indirect effects on mRNA metabolism. More recent publications have found that m6A acts as a repressive mark both in terms of nuclear export (Smalec et al, 2022 Preprint; Tang et al, 2024), and RNA stability (Liu et al, 2020). m6A may even act as a feedback signal to silence chromatin regions that produce transposable element-derived RNA (Chen et al, 2021; Liu et al, 2021). One of the reports that linked m6A to mRNA nuclear export also observed that upon the co-depletion of METTL3 and METTL14 there was a large increase in the cytoplasmic accumulation of m6A-modified mRNAs, suggesting that m6A actually represses nuclear

and standard error of at least three independent experiments, each experiment consisting of at least 30–60 cells. *t* test was performed, \*\*\*P < 0.001. **(E, F)** U2OS cells were transfected with the indicated *ftz* reporters. 6 h post-transfection, cells were treated with 100 µM STM2457 or DMSO. 18 h later, the cells were fixed, and the mRNA was visualized by FISH. Representative images are shown in (E), scale bar = 20 µm, and quantification shown in (F). Each bar represents the average and standard error of at least three independent experiments, each experiment consisting of at least 30–60 cells. *t* test was performed, \*P < 0.05. **(G, H, I)** U2OS cells were treated with lentivirus containing four different shRNAs against METTL3 or scrambled control shRNA. **(G)** Lysates were collected after 96 h, separated by SDS–PAGE, and immunoprobed for METTL3 or tubulin. **(H, I)** Control- or METTL3-depleted cells were transfected with the indicated *ftz* reporters. 18–24 h later the cells were fixed and the mRNA was visualized by FISH. Representative images are shown in (H), scale bar = 20 µm, and quantification is shown in (I). Each bar represents the average and standard error of three independent experiments, each experiment consisting of at least 30–60 cells. *t* test was performed, \*\*P < 0.01.
Source data are available for this figure.

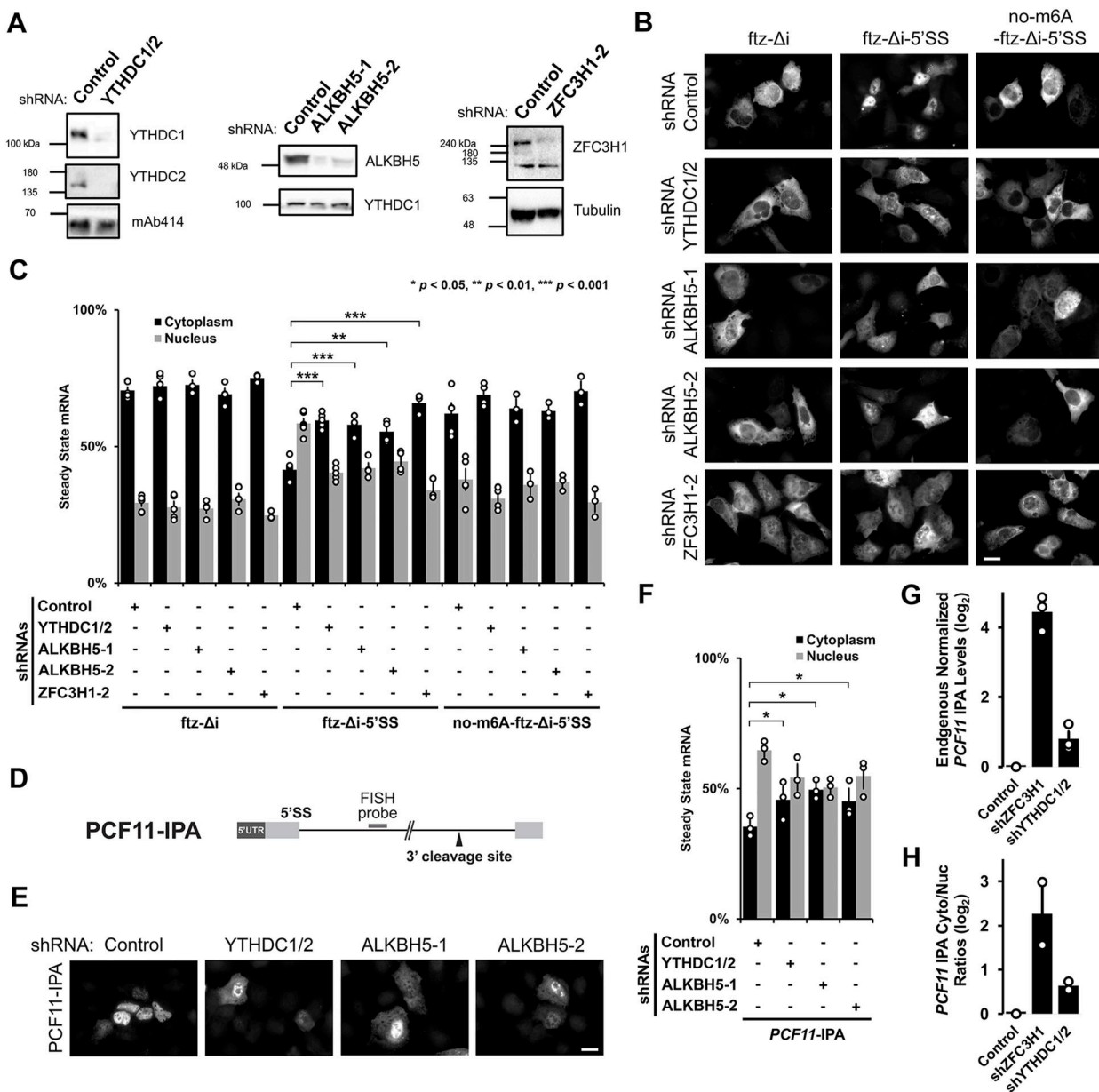

**Figure 3. YTHDC1/YTHDC2 and ALKBH5 are required for the nuclear retention of 5′SS motif containing mRNAs.**
**(A)** U2OS cells were treated with different lentivirus shRNAs against YTHDC1, YTHDC2, ALKBH5, ZFC3H1, or scrambled control. Note that "ZFC3H1-2" denotes shRNA #2 against ZFC3H1 as described previously (Lee et al, 2022). Lysates were collected after 96 h, separated by SDS–PAGE and immunoprobed for YTHDC1, YTHDC2, ALKBH5, ZFC3H1, tubulin, or mAb414, which recognizes FG-nucleoporin proteins. To effectively deplete YTHDC1 and YTHDC2, cells were treated with lentivirus containing two shRNAs against YTHDC1 and another two against YTHDC2. **(B, C)** Control- or YTHDC1/2-, ALKBH5-, ZFC3H1-depleted U2OS cells were transfected with the intronless *ftz* reporter plasmid (±5′SS) and 5′SS motif containing reporter that lacks m6A methylation sites (*no-m6A-ftz-Δi-5′SS*). 18–24 h later the cells were fixed and the mRNA was visualized by FISH. Representative images are shown in (B), scale bar = 20 *μm*, and quantification is shown in (C). Each bar represents the average and standard error of three independent experiments, each experiment consisting of at least 30–60 cells. *t* test was performed, *P < 0.05, **P < 0.01, ***P < 0.001. **(D)** Schematic of *PCF11-IPA* RNA reporter. The position of the FISH probe used to visualize the IPA transcript is marked in gray and the position of the 3′ cleavage site in the intron is as indicated. **(E, F)** Similar to (B, C), except that Control-, YTHDC1/YTHDC2-, or ALKBH5-depleted cells were transfected with the *PCF11-IPA* RNA reporter. **(G, H)** Total RNA was isolated from whole cell lysates (G) or nuclear and cytoplasmic fractions (H) from control- or YTHDC1/2-, ZFC3H1-depleted U2OS cells. *PCF11*-IPA was assessed by qRT–PCR using oligo-dT to generate cDNA and intronic-specific primers for the amplicon. Levels were normalized to the control depletion. Note that no signal was detected in reactions lacking reverse transcriptase (not shown). Each bar represents the average and standard error of three (G) or two (H) independent experiments.
Source data are available for this figure.

mRNA export (Roundtree et al, 2017). Other studies have found that when m6A-modified mRNAs are in the cytoplasm, they are targeted for decay by UPF1 and the CCR4-NOT deadenylase complex, and this

may represent a fail-safe mechanism to ensure the silencing of any m6A-enriched RNA that leaks out of the nucleus (Wang et al, 2014; Du et al, 2016; Boo et al, 2022).

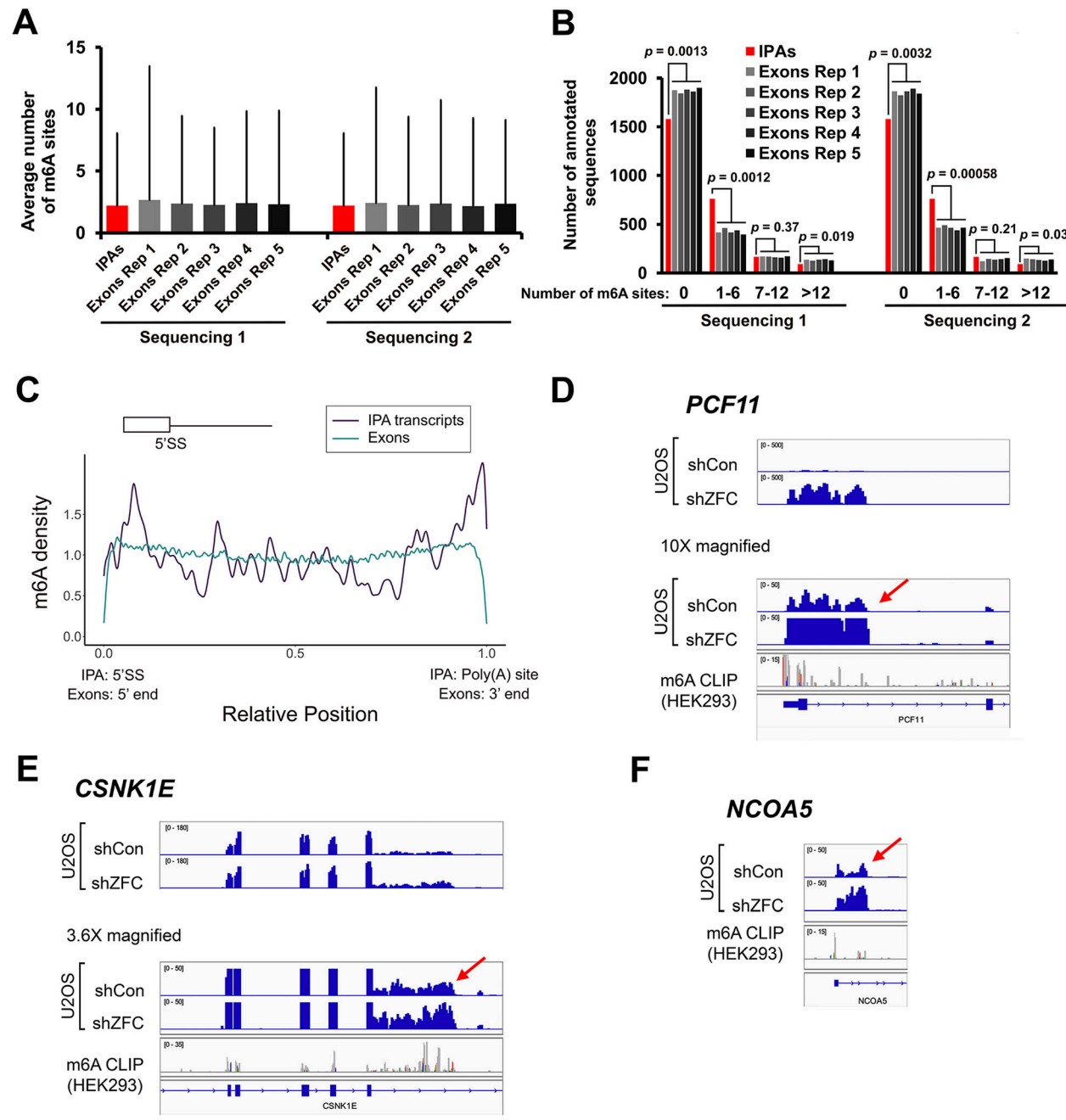

**Figure 4. m6A methylation marks accumulate in IPA transcripts.**
**(A, B)** m6A deposition was assessed by m6A-miCLIP RNA-seq from HEK293 cells. For each intronic region that generates an IPA, an exon that was length (±10%) and expression (±10%) matched was randomly selected. This procedure was repeated five times to generate six sets (IPAs and exons Rep 1–5). The six sets were then compared for the average number and SD of m6A sites (A) or by stratifying them based on the number of sites (B). To ensure reproducibility, the relative expression of intronic regions that generate IPAs and exons were determined based on two independent RNA-seq experiments ("Sequencing 1" and "Sequencing 2"). **(C)** Meta-analysis of m6A density (as determined by m6A-miCLIP RNA-seq) along intronic portions of IPA transcripts (dark blue) and all exons (cyan). The position of the 5′SS motif and poly(A) site is indicated. In IPA transcripts, m6A methylation marks are enriched near the 5′SS motif and poly(A) sites. In exons, m6A methylation marks are depleted from the 5′ and 3′ ends (i.e., near exon-exon junctions), as documented by others. **(D, E, F)** Three examples of IPA transcripts that accumulate m6A marks. IGV browser tracks are shown for miCLIP density. Intronic cleavage/polyadenylation sites are indicated by the red arrows. Total RNA sequencing levels in U2OS cells from previously published work (Lee et al, 2022) are shown. Note that the levels of IPA transcripts are up-regulated after ZFC3H1 depletion (shZFC).

One unresolved question is why the depletion of ALKBH5 promoted the export of IPA transcripts. One possible explanation is that ALKBH5-depletion disrupted nuclear speckles, as previously reported (Zheng et al, 2013), which are required for the retention of reporter mRNAs with intact 5′SS motifs (Lee et al, 2022). When we stain ALKBH5-depleted U2OS cells for the nuclear speckle marker

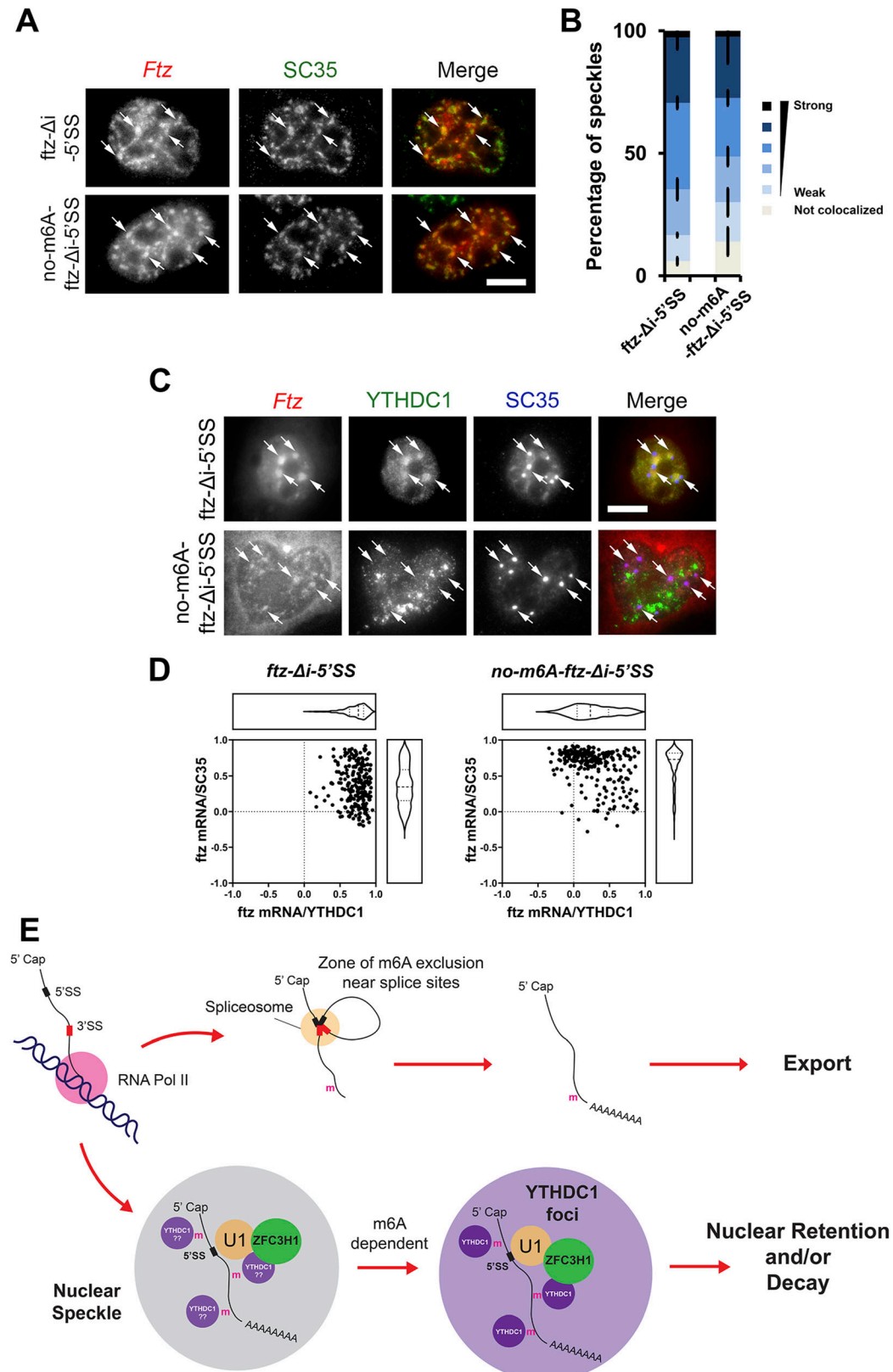

**Figure 5. 5'SS motif containing mRNAs are transferred from nuclear speckles to YTHDC1-containing foci in an m6A-dependent manner.**
**(A, B)** Plasmid DNA containing the *ftz-Δi-5'SS* or *no-m6A-ftz-Δi-5'SS* reporter genes was microinjected into U2OS nuclei. One hour post-injection, cells were fixed and stained for *ftz* mRNA by FISH and SC35 by immunofluorescence. Representative cells are shown in (A), each row represents a single field of view. The merged image shows

SC35, we do not see any obvious changes in nuclear speckles (data not shown). Nevertheless, it is possible that ALKBH5-depletion alters some property of speckles that is not obvious by gross inspection of SC35 immunofluorescence. Another possible explanation is that there is some compensatory effect in ALKBH5-depleted cells by the other demethylases (FTO and perhaps ALKBH3), but this needs to be tested.

Finally, our results strongly indicate that links between the m6A recognition proteins (i.e., YTH domain–containing proteins) and the PAXT complex are evolutionarily conserved. In both humans and *S. pombe*, YTH domain–containing proteins appear to be involved in suppressing transposable element-derived RNAs, suggesting additional conserved features. In *S. pombe*, both the MTREC complex and Mmi1 act together to degrade meiotic mRNAs that are inappropriately expressed in interphase cells. MTREC and Mmi1 are bridged by Erh1, which has a human homolog, ERH, that has been recently shown to regulate the stability of transposable element-derived RNAs and meiotic mRNAs (McCarthy et al, 2021). Thus, it is likely that many aspects of the MTREC-Mmi1 pathway in *S. pombe* are conserved in humans, and our latest results further confirm this idea. Future experiments need to be performed to determine whether other aspects of MTREC and Mmi1 function, including their regulation by TOR signaling and their ability to stimulate heterochromatin formation (Vo et al, 2019; Wei et al, 2021) are conserved in human cells.

# Materials and Methods

### Plasmids constructs and antibodies

All plasmids for reporter constructs in pcDNA3.0, pcDNA3.1 (V5-His), or pcDNA5 are as previously described (Palazzo et al, 2007; Akef et al, 2013; Lee et al, 2015, 2020, 2022). For full length FLAG-ZFC3H1-HA, an insert within the previously published plasmid (FLAG-ZFC3H1 [1–1,233]) was deleted by PCR. The *no-m6A-ftz* and *e-m6A-ftz* reporters were synthesized and inserted in pUC57 plasmids (Genscript), subcloned using HindIII and SmaI sites and inserted into pcDNA3.0 or pcDNA3.1 (V5-His), respectively. The FLAG-HA-tagged YTHDC1 plasmid was purchased from Addgene (plasmid #85167).

Antibodies used in this study include rabbit polyclonals against FLAG (F7425; Sigma-Aldrich), METTL3 (A8370; ABclonal), YTHDC1 (ab122340; Abcam), YTHDC2 (ab176846; Abcam), ALKBH5 (HPA007196; Atlas Antibodies), ZFC3H1 (A301-457A; Bethyl laboratories), U1-70K (ab83306; Abcam), ALY (Zhou et al, 2000), G3BP2 (16276-1-AP; Proteintech), HA (SAB4300; Sigma-Aldrich), Lamin B1 (ab16048; Abcam), eIF4AIII (generous gift from Dr. Lykke-Anderson's laboratory), TRAP-α (Görlich et al, 1990), and mouse monoclonals against SC35 (Clone

35; Sigma-Aldrich), FG-nucleoporins (mAb414; Sigma-Aldrich), and α-tubulin (DM1A; Sigma-Aldrich). Immunofluorescence was performed with Alexa647-conjugated secondaries (Thermo Fisher Scientific) and HRP-conjugated secondaries (Cell Signaling). All antibodies were diluted 1:1,000 to 1:2,000 for Western blotting.

### Cell culture, DNA transfection, cell fractionation, and METTL3 inhibition by STM2457

U2OS and HEK293T cells were grown in DMEM media (Wisent) supplemented with 10% FBS (Wisent) and 5% penicillin/streptomycin (Wisent). DNA transfection experiments were performed as previously described (Palazzo et al, 2007; Akef et al, 2013; Lee et al, 2015, 2020, 2022). U2OS and HEK cells were transfected with the appropriate amount of DNA plasmid according to the manufacturer's protocol using GenJet U2OS or Lipo293T DNA in vitro transfection reagent (SignaGen Laboratories) for 18–24 h.

Cell fractionation was performed as previously described (Lee et al, 2020). Equivalent amounts of nuclear and cytoplasmic lysates were analysed.

For METTL3 inhibition by STM2457, U2OS cells were transfected with the appropriate plasmids as described above. 6 h post-transfection, cells were treated with 100 $\mu$M STM2457 (Sigma-Aldrich) or in DMSO for 18 h. Cells were fixed and FISH was performed as described below.

### FISH staining, immunostaining, and analysis of nuclear speckle targeting

FISH and immunostaining were performed as previously described (Gueroussov et al, 2010; Lee et al, 2015, 2020, 2022). Analysis of nuclear speckle targeting (Fig 5B) was performed as previously described (Akef et al, 2013). Each analysed region of interest consists of a rectangular region drawn to cover a single SC35-positive speckle and its surroundings (1–4 $\mu$m$^2$). Note that individual nuclear speckles were categorized using Pearson correlation intervals of 0.1, from strongest (Pearson correlation of 1 to 0.9) to weakest (Pearson correlation of 0.6 to 0.5). Any Pearson correlation < 0.5 was scored as "not localized." Analysis of RNA foci (Fig 5D) was similarly performed except that each analysed region of interest consists of a rectangular region drawn to cover an RNA foci and its surroundings (1–4 $\mu$m$^2$) and Pearson correlation between the RNA and either SC35 or YTHDC1 was tabulated.

### Lentiviral-delivered shRNA protein depletion

The lentiviral-delivered shRNA protein depletion was performed as previously described (Palazzo et al, 2007; Akef et al, 2013; Lee et al,

---

mRNA in red and SC35 in green. Examples of speckles containing mRNA are indicated by arrows and the scale bar = 10 $\mu$m. Individual nuclear speckles (as detected by SC35) were analysed for colocalization with *ftz* mRNA by Pearson's Correlation Analysis (B) as previously described (Akef et al, 2013). Each bar is the average and standard error of the mean of three experiments; each experiment consists of 100 nuclear speckles analysed from 10 cells. **(C, D)** Plasmid DNA containing the *ftz-Δi-5'SS* or *no-m6A-ftz-Δi-5'SS* reporter genes was transfected into U2OS. 24 h post-transfection, cells were fixed and stained for *ftz* mRNA by FISH and for YTHDC1 and SC35 by immunofluorescence. Representative cells are shown in (C); each row represents a single field of view. The merged image shows mRNA in red, YTHDC1 in green and SC35 in blue. *Ftz* mRNA foci are indicated by arrows and the scale bar = 10 $\mu$m. Individual *ftz* mRNA foci were analysed for colocalization with either SC35 (y-axis) or YTHDC1 (x-axis) by Pearson's Correlation Analysis (D). Each dot represents a single mRNA foci (each cell containing 5–10 foci, cells were obtained from two independent experiments). **(E)** Model of how m6A modifications are required for the nuclear surveillance of 5'SS motif containing mRNAs (see text for more details).

2015, 2020, 2022). HEK293T was plated at 50% confluency on 60 mm dishes and transiently transfected with the gene specific shRNA pLKO.1 plasmid (Sigma-Aldrich), packaging plasmid (Δ8.9) and envelope (VSVG) vectors using Lipo293T DNA in vitro transfection reagent (SignaGen Laboratories) according to the manufacture's protocol. Viruses were harvested from the media 48 h post-transfection and added to U2OS cells pre-treated with 8 µg/ml hexadimethrine bromide. Cells were selected with 2 µg/ml puromycin media for at least 4–6 d. Western blotting was used to determine the efficiency of YTHDC1/YTHDC2, METTL3, ALKBH5, and ZFC3H1 depletion. The shRNA constructs (Sigma-Aldrich) used in this study are as follows: ZFC3H1-2 "TRCN0000432333" 5'-CCGGGACTGATGACATCGCTAATTTCTCGAGAAATTAGCGATGTCATCAGTC-TTTTTTG-3', YTHDC1-1 "TRCN0000243987" 5'-CCGGTGGATTTGCAGGCGTGAATTACTCGAGTAATTCACGCCTGCAAATCCATTTTTG-3', YTHDC1-2 "TRCN0000243988" 5'-CCGGCACCAGAGACCAGGGTATTTACTCGAGTAAATACCCTGGTCTCTGGTGTTTTTG-3', YTHDC2-1 "TRCN0000157638" 5'-CCGGCGGAAGCTAAATCGAGCCTTTCTCGAGAAAGGCTCGATTTAGCTTCCGTTTTTTG-3', YTHDC2-2 "TRCN0000232759" 5'-CCGGCAGAAGTGGCATAGCTTATTTCTCGAGAAATAAGCTATGCCACTTCTGTTTTTG-3', ALKBH5-1 "TRCN0000064783" 5'-CCGGGAAAGGCTGTTGGCATCAATACTCGAGTATTGATGCCAACAGCCTTTCTTTTTG-3', ALKBH5-2 "TRCN0000064787" 5'-CCGGCCTCAGGAAGACAAGATTAGACTCGAGTCTAATCTTGTCTTCCTGAGGTTTTTG-3', METTL3-1 "TRCN0000034714" 5'-CCGGGCCTTAACATTGCCCACTGATCTCGAGATCAGTGGGCAATGTTAAGGCTTTTTG-3', METTL3-2 "TRCN0000034717" 5'-CCGGGCCAAGGAACAATCCATTGTTCTCGAGAACAATGGATTGTTCCTTGGCTTTTTG-3', METTL3-3 "TRCN0000289743" 5'-CCGGGCAAGTATGTTCACTATGAAACTCGAGTTTCATAGTGAACATACTTGCTTTTTG-3', METTL3-4 "TRCN0000289812" 5'-CCGGCGTCAGTATCTTGGGCAAGTTCTCGAGAACTTGCCCAAGATACTGACGTTTTTG-3'.

**FLAG co-immunoprecipitation**

FLAG co-immunoprecipitation was performed as previously described (Lee et al, 2022), except with these modifications. 1 × 150 mm dish of HEK293 cells were transfected with either FLAG-HA-YTHDC1, FLAG-ZFC3H1-HA, or FLAG-ZFC3H1 (1–1,233) using Lipo293T DNA in vitro transfection reagent according to manufacturer's instructions (SignaGen Laboratories). 18–24 h post-transfection the cell pellets were trypsinized, collected, and washed with three times with in ice-cold 1X PBS and flash frozen in liquid nitrogen. In parallel, another 150 mm dish of untransfected HEK293 cells of similar confluency was collected. The pellets were lysed in 2 ml lysis buffer (20 mM Tris–HCl, pH 8, 137 mM NaCl, 1% NP-40, 2 mM EDTA, and *cOmplete* mini-protease inhibitor [Roche]) for 10 min rocking on a nutator at 4°C. To ensure that the nuclear proteins were released from chromatin, the lysed sample was passed through a 25¾'' needle 5X times to shear the chromatin before incubation. The lysate was cleared by spinning at 16,100g for 10 min ~950 µl of the lysate was added to ~50–60 µl FLAG M2 beads (A2220; Sigma-Aldrich), previously preincubated overnight with 5% BSA and 667 µg/ml Ultra-Pure salmon sperm DNA (Invitrogen), and incubated for 3 h or overnight at 4°C. The remaining ~50 µl of the lysate was reserved as "INPUT," and 5X sample buffer was added, and the sample was boiled at 95°C for 5 min. After incubation, the FLAG M2 beads were washed two to five times with 300 mM wash buffer (20 mM Tris–HCl, pH 8, 300 mM NaCl, 0.05% NP-40, 2 mM EDTA, and *cOmplete* mini-protease inhibitor [Roche]), one time with 137 mM wash buffer (same as for 300 mM wash buffer, except with 137 mM NaCl) and transferred to a new Eppendorf tube. Subsequently, for RNase digested samples, ~400 µl of 1:1,000 RNase A/T1 (EN055; Thermo Fisher Scientific) diluted in 137 mM wash was added to the FLAG M2 beads and incubated for 20 min at 4°C. Samples that were not RNase digested were treated in the same manner, except the RNase A/T1 was omitted in the incubation mixture. The samples were washed twice with 137 mM wash buffer, transferred to a new Eppendorf tube and 50–100 µl 2.5X sample buffer was added to sample and boiled at 95°C for 5 min. Samples were separated on an SDS–PAGE gel and transferred onto a blot for immunoblotting.

**RNaseA/T1 treatment**

The RNase efficacy in our co-immunoprecipitation experiments was validated by showing the loss of RNA integrity upon RNase treatment using a denaturing agarose gel. The co-immunoprecipitate lysate was treated with or without RNaseA/T1 followed by RNA extraction by Trizol extraction, followed by isopropanol precipitation. RNA samples were denatured using formamide and formaldehyde in MOPS buffer and incubated at 65°C for 30 min. Denatured RNA was run on an agarose gel supplemented with formaldehyde and stained with SYBR Gold. Integrity was assessed by visualizing the presence of intact 28S and 18S rRNA bands in the gel.

**m6A-miCLIP-seq**

miCLIP-seq was performed as previously described (Linder et al, 2015), with modifications as detailed below. Briefly, cells were grown in independent batches to represent biological replicates. We extracted total RNA from HEK293 cells using TRIzol according to manufacturer's instructions. After RNA was treated with DNase I, 12 µg of total RNA was used for miCLIP-seq. 2 µl of RNA Fragmentation Reagents (AM8740; Invitrogen) was added to the RNA suspended in 20 µl of sterile water. The mixture was incubated in a heat block at 75°C for exactly 12 min. Reaction was stopped by adding 2 µl of stop solution (AM8740; Invitrogen). The fragmented RNA was incubated with 10 µg of m6A antibody (RN131P; MBL) for 2 h at 4°C. RNA-antibody complexes were UV crosslinked twice with 0.15 J/cm² at 254 nm in a Stratalinker 1800. 100 µl of Protein G Dynabeads were added, and the samples were rotated for 2 h at 4°C. Samples were washed three times using iCLIP high salt buffer with 5 min rotation in the cold room for each wash. RNA dephosphorylation was performed using FastAP and T4 polynucleotide kinase. Pre-adenylated L3 adaptors were ligated to the 3'ends of RNAs using the enhanced CLIP ligation method, as detailed previously (Nabeel-Shah & Greenblatt, 2023). RNA was 5'-end-labeled with ³²P using T4 polynucleotide kinase (New England Biolabs, catalogue number M0201L). Samples were separated using 4–12% BisTris-PAGE and transferred to a nitrocellulose membrane (Protran). RNA was recovered from the membrane by digesting proteins with proteinase K (Thermo Fisher Scientific, catalogue number 25530049). Recovered RNA was reverse transcribed into cDNA using barcoded iCLIP primers. The cDNA was size-selected (low: 70–85 nt, middle: 85–110 nt, and high: 110–180 nt) using 6% TBE-Urea gels. cDNA was circularized using CircLigase II ssDNA ligase to add the

adaptor to the 5′-end. We added betaine at a final concentration of 1 M in the reaction, and the mixture was incubated for 2 h at 60°C, as detailed previously (Nabeel-Shah & Greenblatt, 2023). Circularized cDNA was digested at the internal BamHI site for linearization. Linearized cDNA was PCR amplified using Phusion High-Fidelity PCR Master Mix (M0531S; NEB).

The final PCR libraries were agarose gel purified on purification columns (QIAGEN). The eluted DNA was mixed at a ratio of 1:5:5 from the low, middle, and high fractions and submitted for sequencing on an Illumina NextSeq 500 platform to obtain single-end 51 nucleotide reads with 40 million read depth per sample. miCLIP-seq data are deposited at Gene Expression Omnibus (GEO, https://www.ncbi.nlm.nih.gov/geo/) with the accession code GSE230846 for the series, which contains several other datasets that have been recently published (Nabeel-Shah et al, 2024).

### miCLIP-seq analysis

miCLIP-seq analyses were performed as detailed in our previous reports for iCLIP-seq (Song et al, 2022). Briefly, 51-nt miCLIP-seq raw reads that consist of three random positions, a 4-nt multiplexing barcode, and another two random positions, followed by the cDNA sequence were de-duplicated based on the first 45 nt, and random positions, barcodes, and any 3′-bases matching Illumina adaptors were removed. Reads shorter than 25 nt were discarded, and the remaining reads were trimmed to 35 nt using Trimmomatic (Bolger et al, 2014). Reads were mapped to the human genome/transcriptome (Ensembl annotation hg19) using Tophat (Trapnell et al, 2009) with default settings where reads with a mapping quality < 3 were removed from further analysis, which also removes multi-mapping reads.

### Crosslinked-induced truncation sites (CITS)

The precise m6A site was inferred by identifying m6A antibody CITS because of the fact that reverse transcriptase does not efficiently transcribe across crosslinked regions. CLIP Tool Kit was used to identify RNA-binding sites at single-nucleotide resolution (Shah et al, 2017). As miCLIP-seq aims to capture the CITS during cDNA preparation, we therefore called CITS, rather than CIMS (crosslink-induced mutations), on individual replicates. CITS with FDR ≤ 0.01 were considered as significant. CITS from two replicates were merged at a maximum gap of 1 bp. Each peak was then extended 2 bp both upstream and downstream of the identified CITS. Subsequently, only those CITS with a DRACH motif were selected and the coordinates of the base "A" in the motif were designated as m6A sites.

### Bioinformatic analysis of m6A density and distribution in IPA transcripts

A list of introns that do not overlap with exons on the opposite strand was compiled from Gencode v.19. To identify ZFC3H1-regulated IPA transcripts, intronic reads from shZFC3H1 U2OS Frac-Seq data (Lee et al, 2022) were analysed using rtracklayer (Lawrence et al, 2009) to identify introns with enrichment in the first

20% over the last 20%. Next, introns with more overall reads in shZFC3H1 compared with shControl conditions were selected. IPA 5′ splice sites were inferred from the PolyA feature annotation file from Gencode.

To estimate the levels of IPA transcripts compared with exons in HEK293 cells, previously published RNA-seq data (GEO: GSE165772) was aligned to the Ensemble GRCh38 genome by STAR (Dobin et al, 2013). Samtools (Danecek et al, 2021) coverage command was used to obtain the RNA-seq coverage (meandepth column) at all the unique IPA transcripts and exon sites. To compare relative m6A levels, each IPA transcript was randomly matched with an exon that was both size matched (in nucleotide length, allowing for ±10% difference) and expression-matched (mean read depth, allowing for ±10% difference). Average m6A reads per IPA transcripts or exon were compiled. The IPA transcript-exon matching procedure was repeated five times, to determine whether differences were statistically significant. This entire procedure was then repeated using two separate RNA-seq runs (GSM5049715 and GSM5049716). It should be noted that many IPAs are present at extremely low levels and this likely impeded our ability to detect m6A sites; however, this is equally true for the expression-matched exon group.

To determine the distribution of m6A reads from the miCLIP analysis along IPA transcripts and exonic regions, m6A peaks were aligned to IPAs or exons using their genomic coordinates and normalized to the length of the IPA or exon. The distribution was plotted using the ggplot2 density geometry (Wickham, 2009). Most bioinformatical analyses were performed using R Statistical Software v4.2.2 (R Core Team, 2022).

## Data Availability

miCLIP-seq data are available at Gene Expression Omnibus (GEO, https://www.ncbi.nlm.nih.gov/geo/) with the accession code GSE230846 for the series, which contains several other datasets that have been recently published (Nabeel-Shah et al, 2024).

## Supplementary Information

## Acknowledgements

We thank members of the Donnelly Centre RNA sequencing facility for performing the m6A-miCLIP RNA sequencing. We thank GL Burke and N Ahmed for assistance with the m6A-miCLIP library preparation, J Gao for her helpful discussions for the bioinformatics analysis, and WS Tse for his technical expertise for our computational needs. Finally, we thank members of the Palazzo laboratory for their helpful discussions and insights. This work was supported by a grant to AF Palazzo from the Natural Sciences and Engineering Research Council of Canada (FN 492860).

## Author Contributions

ES Lee: conceptualization, investigation, and writing—original draft, review, and editing.
HW Smith: investigation.
YE Wang: investigation.
SSJ Ihn: investigation.
L Scalize de Oliveira: investigation.
NS Kejiou: investigation.
YL Liang: investigation.
S Nabeel-Shah: investigation.
RY Jomphe: data curation and investigation.
S Pu: data curation and investigation.
JF Greenblatt: resources and supervision.
AF Palazzo: conceptualization, resources, supervision, funding acquisition, investigation, project administration, and writing—original draft, review, and editing.

## Conflict of Interest Statement

The authors declare that they have no conflict of interest.

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
