## [Reviewer comments · Life Science Alliance]

Life Science Alliance

N-6-methyladenosine (m6A) Promotes the Nuclear Retention of mRNAs with Intact 5' Splice Site Motifs

Eliza Lee, Harrison Smith, Yifan Wang, Sean Ihn, Leticia Scalize de Oliveira, Nevraj Kejjou, Yijing Liang, Syed Nabeel-Shah, Robert Jomphe, Shuye Pu, Jack Greenblatt, and Alexander Palazzo

DOI: <https://doi.org/10.26508/lsa.202403142>

Corresponding author(s): Alexander Palazzo, University of Toronto

Review Timeline:	Submission Date:	2024-11-19
	Editorial Decision:	2024-11-20
	Revision Received:	2024-11-20
	Accepted:	2024-11-22

Transaction Report:

Please note that the manuscript was reviewed at Review Commons and these reports were taken into account in the decision-making process at Life Science Alliance.

Please note that the manuscript was previously reviewed at another journal and the reports were taken into account in the decision-making process at Life Science Alliance.

Manuscript number: RC-2023-02055

Corresponding author(s): Alexander Palazzo

1. General Statements

In this manuscript, we provide the first evidence that the N⁶-methyladenosine (m⁶A) modification and two m⁶A-binding proteins, YTHDC1 and YTHDC2, are required for the nuclear retention of misprocessed mRNAs in mammalian cells, specifically intronic polyadenylated (IPA) transcripts which are generated from spurious 3' cleavage in intronic sequences and have intact 5' splice site (5'SS) motifs. We demonstrate that YTHDC1 interacts with ZFC3H1, a component of the Poly(A) Exosome Targeting (PAXT) complex, and U1-70k, a component of the U1 snRNP. Both ZFC3H1 and U1-70k were previously shown to be required for the nuclear retention of these misprocessed mRNAs (Lee et al., *RNA* 2022). Importantly, it was previously shown that the budding yeast homologues of ZFC3H1 (Red1) and YTH proteins (Mmi1) interact with each other and are both required for the nuclear retention of unspliced mRNAs and RNAs from transposable elements. Our results are one of the first indications that this pathway is evolutionarily conserved and that m⁶A may play a general role in mRNA quality control throughout eukaryotes.

All three reviewers agreed that the work was an important advance. Reviewer 1 states "The concept introduced by this paper is exciting and novel." and "The study is highly significant." Reviewer 2 states "These results are important because they add to our understanding of how m⁶A modifications can contribute to post-transcriptional regulation." and "This work also adds to our understanding of the biological consequences of m⁶A modification, which is an area of significant interest." Reviewer 3 states "This study advances our understanding of the quality control of misprocessed transcripts in higher eukaryotes."

2. Description of the planned revisions

Here we address the reviewer's comments, point by point. The reviewer comments are in black and our reply is in red. We number all of the experiments that either have already been performed or are in progress.

Reviewer #1 (Evidence, reproducibility and clarity (Required)):

The concept introduced by this paper is exciting and novel. However, the current paucity of presented data can lead to incorrect interpretations of the findings and speculations that might not hold true after a more rigorous assessment of the observed phenomenon.

The premise of this study builds upon an interaction between the PAXT complex and

Revision Plan

nuclear YTH domain containing proteins. However, figures 1B and C should be improved. The interacting band for the ZFC3H1 presented in panel B does not seem to match the size of the construct used in panel C. Is the Flag version of ZFC3H1 expressing a smaller isoform for this protein?

The reviewer is correct in that endogenous ZFC3H1 (which migrates at 250kD with a minor band at 150kD, see Figure 1B in the initial manuscript) appears to differ from the FLAG-tagged construct as expressed from a plasmid transfected in HEK293 cells (which migrates as two bands at 180kD and 200kD, see Figure 1C in the initial manuscript). For the endogenous protein, the predicted molecular weight is 226kD and the 250kD band disappears when cells are transduced with lentivirus containing shRNAs against ZFC3H1 (see Figure 4A in the initial manuscript), indicating that it is the correct protein. Both the 250kD endogenous protein (*) and the 200kD overexpressed protein (**) in transfected HEK293 and U2OS cells are detected in immunoblots using anti-ZFC3H1 antibodies (see Figure 1 in this document) indicating that the over-expressed protein is indeed ZFC3H1.

Figure 1. Molecular Weight Size Comparison of Endogenous ZFC3H1 and FLAG-ZFC3H1 (1-1233).

Lysates from HEK293 and U2OS that were either untransfected or transfected with FLAG-ZFC3H1 (1-1233) plasmid. We labelled the bands corresponding to the endogenous ZFC3H1 "*" and FLAG-ZFC3H1 (1-1233) "**".

We have sequenced the plasmid, and discovered that it contains an additional sequence inserted within the middle of the ZFC3H1 cDNA with a premature stop codon. As such, the version of the protein that is expressed from the plasmid only contains amino acids 1-1233 of the endogenous protein and is missing amino acids 1234-1989. The deleted region only contains TPR repeats, and is not known to interact with any of the well characterized interactors of ZFC3H1 (Wang, *Nuc Acid Res* 2021, Figure 3). We have renamed this construct FLAG-ZFC3H1 (1-1233). Given these new considerations, our results are consistent with the idea that the N-terminal portion of ZFC3H1 interacts with U1-70K, YTHDC1 and YTHDC2. We will change the text to reflect this.

We are currently in the process of deleting the small insertion to obtain a plasmid that encodes a full length version of human ZFC3H1. For the final manuscript:

Revision Plan

Experiment #1) We will repeat the co-immunoprecipitation with the full length FLAG-ZFC3H1 to determine whether it interacts with YTHDC1 and YTHDC2. This will take a few weeks.

Also, the YTHDC1-2 interaction in panel C is not as convincing considering the negative controls lane show some degree of binding.

Although the reviewer is correct that there is substantial background binding in the YTHDC1 immunoblot, we disagree with their characterization of the results with the YTHDC2 immunoblot (see Figure 1B-C in the initial manuscript). In the new manuscript we have included:

Experiment #2) A new co-immunoprecipitate of the FLAG-tagged ZFC3H1 (1-1233) from HEK293 cells under more stringent conditions where the background level of YTHDC1 binding to beads is negligible. We have already completed this experiment (see Figure 1D in the revised manuscript).

Additionally, can the authors test if their RNaseA treatment worked?

In the new manuscript we have included:

Experiment #3) A new co-immunoprecipitate of FLAG-YTHDC1 immunoblotted for eIF4AIII from HEK293 cell lysates. We find that without RNase, there is some eIF4AIII in the precipitates but that the levels diminish substantially after RNase A/T1 treatment. We have already completed this experiment (see Figure 1B in the preliminary revised manuscript).

Why do you need 18 hours to observe the nuclear export of your modifiable construct when inhibiting METTL3 in figure 3? Is it possible that your observation is secondary to phenotypes these cells develop as a result of blocking METTL3?

We treated cells for this period of time so that during the expression of the reporter, all of the newly synthesized mRNA is expressed in the absence of m6A methyl transferase activity. For shorter treatment times, it is unclear whether the bulk of the reporter mRNA, which would be synthesized before the treatment, would lose any pre-existing m6A marks, making a negative result hard to interpret. Previously we found that although 50% of intronic polyadenylated (IPA) transcripts from our reporters are rapidly degraded, about 50% are stable and are nuclear retained over extended periods of time (see Lee et al., *PLOS ONE* 2015; <https://journals.plos.org/plosone/article?id=10.1371/journal.pone.0122743> Figure 3B-G). We believe that the bulk of the reporter mRNA that we are visualizing is stable and accumulates in the nucleus. Given that METTL3-depletion inhibits nuclear retention and that versions of the IPA reporter that lack m6A modification motifs are exported, we think that the most likely interpretation of the 18 hour STM2457 treatment experiments is that the lack of methyltransferase activity had a direct effect, rather than an indirect effect, on nuclear retention. We would be open to performing more experiments if the editors insist, however we ordered

Revision Plan

STM2457 four weeks ago and it has yet to arrive from Sigma-Millipore. Performing this experiment may substantially delay our ability to resubmit the manuscript in a timely manner.

Is ALKBH5 nuclear and/or cytoplasmic in the cell system used?

According to The Human Protein Atlas, ALKBH5 is predominantly nuclear in U2OS cells, with some present in the cytoplasm

(<https://www.proteinatlas.org/ENSG00000091542-ALKBH5/subcellular#human>).

In the revised manuscript we have included:

Experiment #4) Data from subcellular fractionation demonstrating that ALKBH5 is present in both the nucleus and cytoplasm that we have already performed (see Figure 4J in the preliminary revised manuscript).

Reviewer #1 (Significance (Required)):

The study is highly significant

Reviewer #2 (Evidence, reproducibility and clarity (Required)):

Summary: In the manuscript by Lee et al. entitled "N-6-methyladenosine (m6A) Promotes the Nuclear Retention of mRNAs with Intact 5'Splice Site Motifs", the authors provide evidence that m6A modifications within specific regions of transcripts can confer nuclear retention. These results are important because they add to our understanding of how m6A modifications can contribute to post-transcriptional regulation. Although the authors do not quite come out and say this, data seem to be accumulating to suggest that the location of the m6A modifications within a given transcript can dictate the functional consequences of those modifications.

We thank the reviewer for pointing this out. We have included a few sentences in the new preliminary revised manuscript pointing out that the location of the m6A modification in IPA transcripts, with respect to intact 5'SS and poly(A) signals, may play a role in promoting nuclear retention.

The current work builds on previous findings from these authors identifying factors critical for retention of intronic polyadenylated (IPA) transcripts. The present study identified m6A modification as one of the signals for the retention of such transcripts. The authors use reporters for their analysis and also examine validated endogenous IPA transcripts. The data presented supports the conclusions albeit they show a surprising finding for one of the m6A erasers, ALKBH5. However, there is some controversy over the mechanism by which ALKBH5 functions and whether the m6A mark is truly

Revision Plan

reversible, so these results may continue to add to this point of view.

Major Comments: One experiment that might add to the argument would be overexpression of Mettl3 as compared to catalytically inactive Mettl3. The prediction would be that the reporter transcript with intact DRACH sequences would be even more retained in the nucleus in a manner that depends on Mettl3 catalytic activity. For some of the data presented, the reporter is already wholly nuclear so no difference could be detected, but in the U2OS cells shown in Figure 2B, it appears that an increase in nuclear localization might be evident. Such an experiment would add an orthogonal approach to demonstrate that the methylation by Mettl3 is required for retention. If such an experiment would work with the endogenous IPA transcripts shown in Figure 4, but these transcripts may already be too nuclear to detect any increase in nuclear retention.

We have performed two experiments that try to address this but they gave negative results:

Experiment #5) We have over-expressed wildtype and a methyl transferase mutant FLAG-METTL3 and assessed the nuclear export/retention of *ftz-Δi-5'SS* mRNA. There was no effect (Figure 2 in this document).

Figure 2. Over-expression of METTL3 does not increase the nuclear retention of *ftz-Δi-5'SS*.

U2OS cells were co-transfected with *ftz-Δi-5'SS* reporter and either FLAG-METTL3 or FLAG-METTL3-D395A, which lacks methyl-transferase activity (Wang, *Mol Cell* 2016). Cells were fixed, stained for *ftz* mRNA by fluorescent in situ hybridization and METTL3 using anti-FLAG antibodies. The nuclear and cytoplasmic distribution of *ftz* mRNA was quantified as described in the manuscript. Note that this is the average of one independent experiment (each bar consisting of the average of at least 50 cells). We plan to repeat this two more times, but we anticipate that these will show the same result.

We could include this negative data as a supplemental figure. We believe that there are two possible reasons for this experimental result. First, as the reviewer points out, the reporter transcripts are already too nuclear to detect any significant change. Second, METTL3 is part of a larger complex that includes several proteins including METTL14, WTAP and potentially other proteins (for example see Covelo-Molares, *Nuc Acid Res* 2021). We may need to co-express all of these proteins to see an effect.

Experiment #6) We have also expressed versions of *ftz-Δi* and *ftz-Δi-5'SS* mRNA with optimized m6A modification (i.e. DRACH) motifs (AGACT) to enhance methylation (“*e-m6A-ftz*”). We only

Revision Plan

observed a slight increase in nuclear retention but it is not significant (see Figure 2A,C in the revised manuscript).

Again, this result could be explained by the fact that the reporter is too nuclear to detect any significant increase in retention. We had originally performed this in parallel with the *no-m6A-ftz-Δi-5'SS* reporters but did not report this negative data in the original manuscript.

Some rather minor changes to the presentation of the data could enhance the impact of this study.

Specific Comments:

The primary question in this manuscript is comparing reporters with m6A site (intact DRACH sequences) to those without. For this reason, organizing the data to the +/- DRACH sites are adjacent to one another might make the most sense. This point is evident in Figure 1C where perhaps simply changing the order of the bars presented to put the ones directly compared adjacent would be preferable. Then the p-value would compare sets of data directly adjacent to one another.

We thank the reviewer for this suggestion and we have made these changes to the figures in the preliminary revised manuscript.

While the authors show representative fields/cells for most assays, they do an excellent job of providing quantitation as well. One exception is Figure 3D, which shows a single cell image for the most key panel (the 5'SS-containing reporter upon Mettl3 depletion). If there is not a field with more cells, the authors could create a montage.

In the revised manuscript, we have replaced this image with one containing multiple cells expressing the reporter.

Minor Comments:

Figure presentation:

The text in a number of the figures is VERY small (Figures 1B,1C, and 4A) for example.

We have fixed this in the new manuscript.

Figure 3A includes the label "shRNA:" at the top, but these cells are treated with Mettl3 inhibitor and there does not appear to be any shRNA employed, so this seems like a labeling error.

Revision Plan

We have fixed this in the new manuscript.

In Figure 3C, the immunoblot of Mettl3, there are three bands that all disappear completely upon knockdown of Mettl3- are these all Mettl3? This should at least be mentioned in the legend and perhaps indicated in the figure. The authors do mention in the text employing shRNAs to target multiple Mettl3 isoforms, so likely this is the case.

We have clarified these issues in the new manuscript.

Minor points (some really minor to just polish the presentation for clarity):

The word "since" should only be used if there is a time element- otherwise the word "as" is preferable.

For example on p. 4, the sentence: "Since inhibition of mRNA export typically enhances the nuclear retention of RNAs with intact 5'SS motifs (Lee et al. 2020),.." would more precisely read "As inhibition of mRNA export typically enhances the nuclear retention of RNAs with intact 5'SS motifs (Lee et al. 2020),..".

We thank the reviewer for pointing this out. We have fixed this issue in the revised manuscript.

Reviewer #2 (Significance (Required)):

Summary: In the manuscript by Lee et al. entitled "N-6-methyladenosine (m6A) Promotes the Nuclear Retention of mRNAs with Intact 5'Splice Site Motifs", the authors provide evidence that m6A modifications within specific regions of transcripts can confer nuclear retention. These results are important because they add to our understanding of how m6A modifications can contribute to post-transcriptional regulation. Although the authors do not quite come out and say this, data seem to be accumulating to suggest that the location of the m6A modifications within a given transcript can dictate the functional consequences of those modifications.

This study would be of significant interest to those that study gene expression in any context as well as cell biologists as the data add to our understanding of export of mRNA from the nucleus. This work also adds to our understanding of the biological consequences of m6A modification, which is an area of significant interest. In my opinion, the authors could make a broader conclusion that we do, which is that the location of the modification significantly dictates function- an extension of previous findings mostly focused on processed mRNA transcripts.

Reviewer #3 (Evidence, reproducibility and clarity (Required)):

Revision Plan

Quality control of mRNA is vital for all types of cells. In eukaryotic cells, nuclear export of misprocessed mRNAs containing the 5' splice site is prevented. In this manuscript, Lee and colleagues demonstrate that the nuclear retention of intronic polyadenylated transcripts is dependent on m6A modification. Based on the results shown in yeast, they perform immunoprecipitation experiments and demonstrate the interaction between ZFC3H1, a component of the PAXT complex, and YTHDC1 and YTHDC 2, nuclear YTH RNA-binding proteins that recognize m6A-modified transcripts. The study also shows the interaction of U1-70K with YTHDC1 and with ZFC3H1. Depletion of YTHDC1/2 prevents the nuclear retention of IPA transcripts. Additionally, CLIP-seq analysis is performed, demonstrating that m6A modification is enriched around the 5' splice site motif and the 3' polyadenylation site in IPAs. From these observations, they conclude that m6A modification contributes to the quality control of mRNA by promoting nuclear retention of misprocessed transcripts.

Major Points

1. The interaction between ZFC3H1 and YTHDC1 is clearly shown by immunoprecipitation of FLAG-tagged YTHDC1 in Figure 1B. However, the co-purification of YTHDC1 with FLAG-tagged ZFC3H1 in Figure 1C is rather ambiguous. Additionally, the immunoprecipitated samples do not appear to show signals corresponding to FLAG-tagged ZFC3H1, making it unclear if the immunoprecipitation is working. It is essential to provide a better quality result to clarify these observations.

Please see our responses to reviewer #1. We have repeated the co-immunoprecipitation of FLAG-ZFC3H1 (1-1233) with YTHDC1 under more stringent conditions and have reduced the background binding (see Figure 1B and D in the new manuscript). We have also determined why the FLAG-ZFC3H1 is smaller than expected as the construct contains a premature stop codon. As explained above, we are in the midst of generating a full-length FLAG-ZFC3H1 and we plan to repeat the co-immunoprecipitation with this new construct.

2. While the authors demonstrate that the m6A modification is dispensable for the targeting of IPA reporter transcripts to the nuclear speckles, it would be valuable to investigate whether m6A is required for their exit from the nuclear speckles. Do reporter transcripts with m6A motifs remain in the nuclear speckles at later time points?

We have now analyzed the colocalization of nuclear speckles (SC35) with *ftz-Δi-5'SS*, which contains both a 5'SS and DRACH motifs, and *no-m6A-ftz-Δi-5'SS*, which contains a 5'SS but lacks DRACH motifs, at steady state – i.e. after 18-24 hours of transfection (as opposed to at early time points as shown in Figure 2D-E of the initial manuscript). Unexpectedly, we see that both mRNAs continue to colocalize with nuclear speckles, although the *no-m6A-ftz-Δi-5'SS* mRNA is well exported from the nucleus and its signal in nuclear speckles is faint (see Figure 2F-H in the new manuscript).

Revision Plan

Previously, we observed that *ftz-Δi-5'SS* required the 5'SS motif to remain in nuclear speckles at these later time points (Lee *PLOS ONE* 2015 and Lee *RNA* 2022). Upon closer inspection, *ftz-Δi-5'SS* mRNA also accumulates in additional nuclear foci that are not SC35-positive. We have data that these other foci are adjacent to nuclear speckles and contain YTHDC1 protein. Our new results may indicate that m6A marks promote the transfer of mRNAs from nuclear speckles to YTHDC1 foci, but more data is required to make a firm statement. Given this, we plan to conduct further experiments which may take a month to complete:

Experiment #7) We are now assessing whether various versions of our reporter (*ftz-Δi-5'SS*) are transferred from nuclear speckles to YTHDC1-positive foci, which were previously shown to partially overlap nuclear speckles and sequester m6A-rich mRNAs (Cheng *Cancer Cell* 2022). Other groups have reported the existence of "pA+ RNA foci" which accumulate MTR4/ZFC3H1-targeted RNAs and ZFC3H1 protein when the nuclear exosome is inhibited (Silla *Cell Reports* 2018) – we suspect that these are the same foci that we see. We plan on co-staining *ftz-Δi*, *ftz-Δi-5'SS*, *no-m6A-ftz-Δi* and *no-m6A-ftz-Δi-5'SS* with SC35, YTHDC1 and ZFC3H1 to determine whether both the presence of 5'SS motifs and m6A are required for the transfer of mRNAs from nuclear speckles to YTHDC1-enriched foci. We will also determine whether these foci contain ZFC3H1. This will take about three weeks to a month to complete.

3. Figures 5B and 5C suggest that ZFC3H1 is required for the degradation of IPA transcripts. However, the range of the vertical axis is inappropriate and it is difficult to assess the extent of the increase in expression levels. Please adjust the vertical axis range for improved clarity.

We thank the reviewer for the feedback we will add additional graphs with an expanded vertical axis to demonstrate that ZFC3H1 is required for the degradation IPA transcripts.

Minor Points

1. page 4, line 2 "RNase" should be corrected to "RNase".

We thank the reviewer for catching this error. We have fixed this.

2. page 7, line 5: Is the statement "prevents the nuclear export and decay of non-functional and misprocessed RNAs" correct? m6A modification promotes the decay of such RNAs.

We thank the reviewer for pointing this out. We have altered the text to clarify that m6A promotes decay.

3. Figure 2E: *ftz-Δi* should be *ftz-Δi-5'SS*.

We thank the reviewer for catching this error. We have fixed this.

4. **Figure 5A:** It would be helpful to indicate the number of IPA transcripts analyzed.

We have included this information.

Reviewer #3 (Significance (Required)):

Overall, the work is sound and generally well-controlled. This study advances our understanding of the quality control of misprocessed transcripts in higher eukaryotes. This reviewer suggests a few points for clarification or improvement.

3. Description of the revisions that have already been incorporated in the transferred manuscript

As detailed above, we have incorporated experiment #2 (Figure 1D), experiment #3 (Figure 1B), experiment #4 (Figure 4J), experiment #6 (Figure 2A-C) into the new manuscript. We have also addressed the other comments by changes in the text and figures, as described above.

Although we have performed a number of experiments, there are a few outstanding experiments we wish to carry out before updating the manuscript and resubmitting it.

- 1) We need to generate a full length FLAG-ZFC3H1 and repeat the co-immunoprecipitation with YTHDC1 and YTHDC2 (experiment #1).
- 2) We need to replicate the METTL3 result (see experiment #5, above), although we do not plan to include this in the final manuscript.
- 3) We plan on co-staining *ftz-Δi*, *ftz-Δi-5'SS*, *no-m6A-ftz-Δi* and *no-m6A-ftz-Δi-5'SS* with SC35, YTHDC1 and ZFC3H1 to determine whether m6A and intact 5'SS motifs help to transfer mRNAs from nuclear speckles to YTHDC1 or ZFC3H1-enriched foci (experiment #7). This last experiment may take a month to complete properly.

4. Description of analyses that authors prefer not to carry out

Reviewer #1 asked us to repeat the STM2457-treatments with shorter incubations. Seeing that we are still waiting for the drug to be delivered from Sigma-Millipore, and that these experiments will likely give us ambiguous results (see discussion above), we would prefer not to perform these. Nevertheless, if the editor insists, we will perform these experiments – with the caveat that the delay in STM2457 shipment may delay the resubmission of the manuscript.

Thank you for your patience. We are pleased to submit a revised version of our manuscript, **“N-6-methyladenosine (m6A) Promotes the Nuclear Retention of mRNAs with Intact 5' Splice Site Motifs”** for publication in this journal as a report.

We now include the following changes:

- 1) We have repeated the co-immunoprecipitations with full-length FLAG-ZFC3H1-HA and find that it co-precipitates with YTHDC1 and U1-70K (Figure 1D-E).
- 2) We demonstrate that RNase treatment eliminates RNA from the lysates used in immunoprecipitation assays (Figure S1A).
- 3) We quantify the levels of over-expressed FLAG-ZFC3H1-HA and FLAG-HA-YTHDC1 protein relative to their endogenous counterparts in lysates used in immunoprecipitation assays (Figure S1B-G). We find that expression is less than an order of magnitude over endogenous levels. We also measure the levels of their interacting partners and these are not drastically affected (Figure S1B-G).
- 4) We demonstrate that the reporter mRNAs are modified by m6A in a DRACH-motif dependent manner (Figure 2B).
- 5) We test whether STM2457 impacts the levels of ZFC3H1, YTHDC1, YTHDC2 and U1-70K (Figure S2). Most are not changed, although YTHDC1 levels go up slightly.
- 6) We test whether YTHDC1/2-depletion and ZFC3H1-depletion have effects on other proteins in the pathway (Figure S3A-B).
- 7) Demonstrate that the co-depletion of YTHDC1 and YTHDC2 inhibits the nuclear retention of an endogenous IPA transcript (*PCF11* IPA; Figure 3G-H).
- 8) We provide additional analysis of the levels of m6A modification in intronic regions that produce endogenous IPA transcripts (Figure 4A-B). We find that fewer IPA transcripts lack m6A modifications when compared to length and expression-matched exons. Conversely, a greater number of IPA transcripts have a low level of m6A modifications than length/expression-matched exons.
- 9) Previously we examined how mRNAs are distributed in the nucleus, one hour after they are synthesized. We now examine mRNA at later time points (Figure 5C-D). We find that mRNAs with intact 5'SS motifs and m6A sites accumulate in YTHDC1-positive foci that are present next to nuclear speckles and that this localization requires m6A modification sites (Figure 5C-D).
- 10) We reformatted the paper so that it is compatible with the requirements of a report in this journal.

What follows is a point-by-point response to your letter and to the additional reviewer's comments. Our responses are in red.

If you have any further questions, please do not hesitate to contact us.

Thank you for submitting your revised manuscript entitled "N-6-methyladenosine (m6A) Promotes the Nuclear Retention of mRNAs with Intact 5' Splice Site Motifs" and for your patience while we sought the views of an additional expert. The comments from this additional reviewer are appended below. After considering the reviews already in place along with the views of this referee, significant concerns unfortunately preclude publication of the current version of the manuscript in this journal.

We believe that we have addressed most of these concerns. We thank the editor and the reviewer as these additions have significantly improved the paper.

As you will see, this reviewer felt that multiple central claims in this work were missing important confirmation and controls. These concerns covered protein interaction data, the role of YTHDC1, imaging data, and the use of reporters, among others.

We have tried our best to answer most of the reviewer's concerns. We detail these below.

Although we disagree with this reviewer that these concerns make this work inappropriate for eventual publication in this journal, we agree that a suitably revised work must address potential offtarget effects of overexpressing the PAXT component ZFC3H1

We now demonstrate that the expression of FLAG-ZFC3H1-HA does not significantly affect endogenous YTHDC1 levels (Figure S1E-G) and expression of FLAG-HA-YTHDC1 does not significantly affect endogenous ZFC3H1 levels (Figure S1B-D).

and must confirm the requirement of YTHDC1, e.g. by knockdown (points 1 and 4).

We had discussed this point with you, and we agreed that we would instead test the degree to which FLAG-HA-YTHDC1 was expressed compared to endogenous YTHDC1. We find that this difference is less than an order of magnitude (Figure S1B-D). We found similar results with FLAG-ZFC3H1-HA (Figure S1E-G).

While we agree with this reviewer that validating the proposed processing of IPA sites in endogenous and not reporter genes would strengthen this work (point 2), we leave this to your discretion.

We now analyze the effects of YTHDC1/2-co-depletion and ZFC3H1-depletion on the endogenous *PCF11* IPA transcript. Either treatment led to an increase in this transcript's levels and an inhibition in its nuclear retention (Figure 3G-H).

We also encourage you to improve m6A-CLIP traces and imaging examples as noted in points 5 and 6.

We now perform additional analyses of m6A mark deposition, and show that fewer IPAs lack m6A modifications when compared to length and expression-matched exons. Conversely, a greater number of IPA transcripts have a low level of m6A modifications than length/expression-matched exons. We also include a third example of an IPA and its m6A deposition profile (*NCOA5*, Figure 4F).

Regarding changes sought by the original three reviewers, we agree that co-IP repeats should be added to validate the interaction data,

We have repeated the co-immunoprecipitations with full-length FLAG-ZFC3H1-HA and find that it co-precipitates with YTHDC1 and U1-70K (Figure 1D-E).

and we concur that a shorter duration of METTL3 inhibition would not be informative.

Finally, we felt that this work is best suited to our Report format and below we include formatting requirements for this article type.

We have now reformatted the article to meet the requirements of a Report in this journal.

Please let us know if you are able to address the major issues outlined above and wish to submit a revised manuscript to this journal.

Reviewer #1 (Comments to the Authors (Required)):

The manuscript by Lee et al. proposes that transcripts with intact 5' splice sites, such as those arising from intronic polyadenylation (IPA), are subject to m6A modification, tagging them for degradation through the binding of YTHDC1/2, PAXT and the nuclear exosome. While this would be of interest, I believe that the quality is insufficient to provide strong support for the author's conclusions. The proposed revision plan is not likely to change this.

Major points

- Fig 1: It took 2 reviewers to alert the authors that they were using a truncated ZFC3H1 protein, missing more than a third of the protein. This does not create trust in the overall meticulousness of the present work.

We are sorry about this. We have repeated the co-immunoprecipitations with full-length FLAG-ZFC3H1-HA and find that it co-precipitates with YTHDC1 and U1-70K (Figure 1D-E).

Also, the choice of ectopic expression in the presence of the endogenous protein (overexpression) may lead to artificial interactions. The authors should examine how much a given factor is overexpressed and at the very least deplete the endogenous un-tagged protein. Endogenous protein tagging would of course be the better option.

We now examine the level of the overexpressed protein in comparison to the endogenous protein (Figure S1B-G). In both cases, these are present at levels that are less than 10 times the endogenous protein level. After discussing this with the editors, we agreed that it was not necessary to deplete the endogenous proteins in these experiments.

Finally, the authors chose eIF4AIII to test for RNaseA treatment quality. While this shows a decrease in the eIF4AIII - YTHDC1 interaction, a significant amount can still be observed. Taken together these interactions are not very convincing.

We now provide a direct assessment of the RNase-treatment by assessing RNA levels with and without RNase treatment (Figure S1A).

- While the authors try to make genome-wide claims, all their results arise from reporter systems, sometimes insufficiently characterized. In particular, a strong emphasis is put on DRACH motives, however, these are highly abundant in the genome and not all subject to modification. A meRIP-qPCR would be needed to confirm that their reporter is actually modified by m6A.

We now provide an analysis of m6A modification on our reporter (Figure 2B). We confirm that it is m6A modified and that this drastically reduced when the majority of the DRACH motifs are mutated.

Also, the analysis of 'endogenous' IPA events at PCF11 and CCDC71 is performed using minigene reporters. This is particularly surprising considering fish probes could have been designed for endogenous transcripts with the possibility to discriminate IPA from mature transcripts using two separate probes.

We now analyze the levels and distribution of the endogenous *PCF11* IPA transcripts using qRT-PCR. We show that its level is elevated and that its cytoplasmic/nuclear distribution is increased after depletion of ZFC3H1 or YTHDC1/2 (Figure 3G-H).

- For the reporter constructs no-m6A-ftz- Δ i-5'SS and e-m6A-ftz- Δ i-5'SS, a DRACH motif is present very close to the 5'SS. The authors state that mutating the DRACH motif did not alter the 5'SS motif AAGGUAAGC. However, it might still affect U1snRNP interaction?

This is possible. In *ftz- Δ i-5'SS* there is a DRACH motif that is 11 nucleotides downstream from the consensus GU and it is changed by 4 nucleotides in the *no-m6A-ftz- Δ i-5'SS*. However, since three out of the four changes are also found in *e-m6A-ftz- Δ i-5'SS*, which is nuclear retained, it is unlikely that these changes affect U1 snRNP interaction.

ftz- Δ i-5'SS: gaaggtaagcctatcccTAACCc

no-m6A-ftz- Δ i-5'SS: gaaggtaagcctatcccAGGCTc

e-m6A-ftz- Δ i-5'SS: gaaggtaagcctatcccAGACTc

Note that the consensus 5'SS GT is underlined, and that the DRACH motif (wild type and mutant forms) is in caps.

The closest upstream DRACH motif is 24 nucleotides away from the consensus GU, but this was not mutated in *no-m6A-ftz- Δ i-5'SS*. Due to space limitations, we do not discuss these issues in the manuscript.

- In order to present endogenous transcript behavior, the authors try to support their claims by showing RNAseq data upon ZFC3H1 depletion (Figure 5). While these data nicely and convincingly show that ZFC3H1 depletion leads to the accumulation of the selected IPA transcripts, no data is provided for YTHDC1. In the absence of available YTHDC1 depletion RNAseq, the authors should at the very least have analyzed IPA transcript levels by qPCR. Perhaps PAXT is acting at IPA endogenous transcripts independently of YTHDC1.

We now analyze the levels and distribution of the endogenous *PCF11* IPA transcripts using qRT-PCR and demonstrate that its levels and distribution are affected by the depletion of ZFC3H1 and the co-depletion of YTHDC1/2 (Figure 3G-H).

- Additionally, in their interpretation of sequencing data (Figure 5B-C), the authors point out that "we found that m6A-CLIP reads map throughout the intronic region that generates an IPA", but generally disregard that at PCF11 the signal in the IPA region is close to background when compared with the exonic signal seen notably at the exon down of the IPA. The statement is also at odds with panel A from the same figure, which places m6A peaks mostly at the 5'ss and the IPA polyA site.

The assessment of m6A sites is very important point. It was very difficult to conduct a fair assessment of the m6A modification levels in IPA transcripts compared to mRNAs, as the former are expressed at much lower levels than the later. We now conduct a more rigorous analysis of this by comparing intronic regions that generate IPA transcripts to length/expression-matched exons. We use two independent RNA seq experiments to determine the relative expression levels of intronic regions that generate IPAs and exons. We find that although the average number of m6A is similar between the two groups, fewer IPA transcripts lack detectable m6A modifications, and more have a low number of sites (1-6) (Figure 4A-B).

As for the distribution of m6A along IPAs, although there is a slight increase at the 5' and 3' ends (Figure 4C in the new version), it is unclear whether this is significant or not. In reality there are m6A marks that map throughout the IPA transcripts (Figure 4C). To determine whether these slight biases are relevant, one could performed additional experiments with reporter genes where select DRACH motifs are mutated. However, we believe that these experiments are beyond the scope of this paper.

- The interpretation of results and displays of immunofluorescence data are also problematic. For instance, in Figure 2 panel D bottom lane, colocalization of the no-m6A-ftz- Δ i-5'SS with nuclear foci is shown. However, the cell selected for the panel shows exclusively nuclear signal, while this construct is reported to be mostly cytoplasmic in panels B and C. Of course, a range of effects are expected but here panels B/C and D are in total opposition.

In these experiments, the reporters were examined only an hour after the onset of their expression, at a time point when they are not yet exported from the nucleus. We now make this clear in the text. We also extend our study to later time points. As can be seen in Figure 5C, no-m6A-ftz- Δ i-5'SS mRNA is predominantly cytoplasmic at these later time points. Despite this, it is still present in the nuclear speckles.

Also, in Figure 2 panel G bottom lane: concluding that there is an actual colocalization between Ftz and SC35 staining, when Ftz staining covers most of the cell (the image was acquired without the use of a confocal microscope) seems erroneous.

The Pearson correlations were conducted on individual RNA foci not on the entire cell. We now make this clear in the text and provide a detailed description of how this analysis was conducted in the Materials and Methods.

- The authors point out themselves that the depletion of ALKBH5 (which removes m6A) displays the same phenotype as the inhibition of m6A through various means (Figure 4) and simply discard this observation through vague discussion statements such as "It is also possible that the depletion of ALKBH5 disrupts some aspect of the m6A metabolism". This constitutes a vague scientific argument.

We have now rewritten this statement.

Minor points

- The authors could improve the display of their data, for instance by simplifying the naming convention of their reporters

These are standard reporters in the field, and we are using the common nomenclature so that individuals can cross reference between publications.

and directly showing the Cytoplasmic/Nuclear ratios.

We report separately the cytoplasmic and nuclear amounts as we believe that this provides a more accurate description. For example if a treatment alters the mRNA distribution from 75% cytoplasmic to 80% cytoplasmic, this is not a substantial change, but reporting it in terms of a ratio (a change of $75/25 = 3$ to $80/20 = 4$) exaggerates the actual change (a supposed increase of export by "25%"). Reporting separate cytoplasmic and nuclear percentage is standard in the mRNA export field.

For RNA-seq experiments, it is hard to give absolute values for nuclear and cytoplasmic RNA levels, as there is no good method to accurately relate these two values. As such we must give ratios for this type of analysis (Figure 3G-H), as we have done in the past (e.g. Lee et al., *NAR* 2020).

- 1.216 to 220 are quite speculative

We now include new data that mRNAs containing intact 5'SS accumulate in YTHDC1 bodies in an m6A-dependent manner (Figure 5C-D). As a result, this section has been extensively revised.

- STM2457 can be acquired from at least 2 independent providers aside from Sigma Millipore and could most likely be obtained from neighboring laboratory should cost be an issue. Delivery time of a relatively common chemical does not seem an adequate reason not to perform an experiment requested by reviewers.

Unfortunately, at the time of submission and of our prior resubmission, STM2457 was on backorder. Since then we did manage to obtain this compound. However, after discussing this with the editor, we felt that further experiments with this compound were not needed as a block in mRNA nuclear export can only be reliably detected after extended periods of time to allow for mRNA nuclear accumulation. Despite this we checked whether treatment with STM2457 affected the levels of select proteins (as you asked below). We now show this data (Figure S2A-B).

- Figure legend: Figure 1. "m6A machinery forms a complex": The figure is not showing any interaction of reader and eraser complexes with ZFC3H1. "YTHDC1/2 nuclear m6A readers interact with ZFC3H1" would match the data better.

We have rewritten the figure title.

- Total protein levels of ZFC3H1 and U1-70K need to be shown for the STM2457 treatment to confirm the absence of indirect effects of the drug.

As stated above, we now show this data (Figure S2A-B).

Attached is our point-by-point response to the last round of comments by the reviewers. Our responses are in red.

Thank you for submitting your revised manuscript entitled "N-6-methyladenosine (m6A) Promotes the Nuclear Retention of mRNAs with Intact 5' Splice Site Motifs". The manuscript has been seen by the three original reviewers whose full comments are appended below. We appreciate your patience during the unusually long review period and subsequently the time needed to arrive at a decision. While the reviewers continue to be overall positive about the work in terms of its suitability for this journal, some important issues remain. Consistent with our prior correspondence, we have also evaluated the evidence for complex formation among ZFC3H1 and YTHDC1/2, in lieu of returning this to the original Reviewer 4 who was very critical of this work.

We thank you for your patience and generosity in allowing us a subsequent round of revisions. I would like to apologise again for the delay in our response. As I pointed out in the last round, the first author is no longer in the lab, and this has delayed us in completing the work in a timely manner.

Here, we find the co-immunoprecipitation gels using full-length proteins, provided in response to the original reviewers in Fig 1, are not convincing. Concerns over protein gels are shared by two of the reviewers below. These include closely cropped gels, contrast settings, as well as very subtle differences in band intensities. These issues reduce confidence in these results, which are key to the main conclusions of this work. A suitably revised manuscript must include quantification (e.g. densitometry) of multiple Western blots from multiple co-IP experiments for

reproducibility and statistical analysis, while adjusting the text to align with data shown if needed.

We have tried to address these issues. We have now conducted the co-immunoprecipitates several times and performed densitometry analysis, which is now displayed in Supplemental Figure 1B-C. After many additional repetitions, we decided to omit the eIF4AIII blot as our densitometry analysis concluded that its level of co-immunoprecipitation with FLAG-HA-YTHDC1 was not significantly reduced after RNase treatment. Indeed, it co-immunoprecipitated with FLAG-HA-YTHDC1 in half of the experiments (with and without RNase) and not in the other half. Despite this, we are confident that the RNase treatment worked as it eliminated RNA from the samples (see Supplemental Figure 1A). We also now provide evidence that YTHDC2 co-immunoprecipitates with full length FLAG-ZFC3H1-HA (Figure 1D) and have repeated this several times and provide statistical analyses of densitometry analysis (Supplemental Figure 1C) to back this up. We also repeated the immunoblot analysis of proteins +/- STM2457 (Figure S2). As for Figure S3 panels F-G, we believe that these appeared over-contrasted due to some pdf conversion problem on the submission site. The version of our previously submitted manuscript that appeared in bioRxiv did not have these issues (see <https://www.biorxiv.org/content/10.1101/2023.06.20.545713v3>). We re-uploaded this figure.

In addition, the multiple METTL3 bands noted by original Reviewer 1 should be clarified in the text.

We now display an alternative blot of METTL3 that displays bands consistent with what has been published in the literature. After re-analyzing multiple samples that we had frozen from the original experiments, we found that the pattern displayed in the older version of the text was not reproducible in the other samples. The pattern shown in the new figure is what we consistently observed in the other experiments and agrees with other published reports (where METTL3 migrates at ~70kDa).

Finally, while we agree with this reviewer that inclusion of FTO in the analysis would make a valuable addition and significantly strengthen these claims, this issue may be addressed with text changes.

In the discussion, we put forward the possibility that FTO (and ALKBH3 a putative m6A demethylase) may play a role in this process.

Please note that reviewers were re-ordered from their order at Review Commons:

Original Reviewer 3: Reviewer 2 below

Original Reviewer 2: Reviewer 3 below

Original Reviewer 1: Reviewer 4 below

Our general policy is that papers are considered through only one revision cycle; however, given that the suggested changes are relatively minor we are open to one additional short round of revision. Please note that we will expect to make a final decision without additional reviewer input upon resubmission but that this will be a full and final critical review carried out by us.

We thank you again for all your time and patience. We have tried our best to answer all of the concerns raised by the reviewers.

Reviewer #2 (Comments to the Authors (Required)):

In this report by Lee et al., the authors propose that transcripts containing 5' splice sites are modified with m6A and retained in the nucleus through the interaction between YTHDC1/2 and ZFC3H1. They demonstrate that m6A modification promotes the transfer of the transcripts from nuclear speckles to YTHDC1-enriched nuclear bodies. The manuscript makes an important contribution to the field of gene expression regulation. Most of the points raised in the first version seems to have been properly addressed. I only have a few minor comments.

We thank the reviewer for their kind assessment of our work.

Specific comments:

The authors provide the co-immunoprecipitation data between full-length ZFC3H1 and YTHDC1. However, the interaction between ZFC3H1 and YTHDC2 is only shown using a truncated ZFC3H1. The heading "YTHDC1 and YTHDC2 interact with both ZFC3H1 and U1-70K" should be toned down.

We now provide evidence that YTHDC2 co-immunoprecipitates with full length FLAG-ZFC3H1-HA (Figure 1D) and have repeated this several times and provide statistical analyses of densitometry analysis (Supplemental Figure 1C) to back this up.

While the authors perform co-depletion of YTHDC1 and YTHDC2, what is the reason for not depleting each one separately?

We have had mixed results with these and could not make firm conclusions, likely because they serve redundant roles. We are currently performing these again and plan to include these experiments in a future manuscript.

The western blots in Figure S3F and S3G appear to be overly enhanced.

We believe that there may have been some issue during the pdf conversion. The version that appeared in bioRxiv did not have these issues (see <https://www.biorxiv.org/content/10.1101/2023.06.20.545713v3>). We re-uploaded this same Supplemental Figure again. We checked, and it appears fine this time.

Reviewer #3 (Comments to the Authors (Required)):

The authors have made extensive revisions to the manuscript to address concerns raised in previous rounds of review.

We thank the reviewer for their kind assessment.

Reviewer #4 (Comments to the Authors (Required)):

This work addresses the role of m6A in the nuclear retention of misprocessed mRNAs. The presented concept is novel and could be important. However, the low quality of some of the data and confounding factors that limit the ability to interpret certain results raise doubt on the validity of the conclusions. Below are points that if clarified/reinforced could improve the story.

1. The interaction between YTHDC1 and ZFC3H1 seems quite weak (especially in figure 1B-C). It is unclear from these results to what extent or what % of endogenous ZFC3H1 molecules might actually operate in complex with YTHDC1.

We now assess the percent of the total ZFC3H1 is in the FLAG-HA-YTHDC1 pulldown and how much YTHDC1 and YTHDC2 are in the FLAG-ZFC3H1-HA pulldowns (Supplemental Figure 1B-C).

2. The claim of RNase treatment perturbing eIF4 binding should be reconsidered as the experiment looks quite dirty and it is hard to quantitate an effect from an n=1.

As described above, after repeating this experiment multiple times, and quantifying the results by densitometry analysis, we feel that these findings (that eIF4AIII co-precipitates with FLAG-HA-YTHDC1 only in the absence of RNase) is not reproducible. In some cases it was present in the immunoprecipitates (with and without RNase), and in some other instances it was absent (with an without RNase). Despite this, we are confident that the RNase treatment worked as it eliminated RNA from the samples (see Supplemental Figure 1A).

3. Figure 2G shows 3 bands for mettl3, one ~100kda and 2 ~70kda (which is close to the predicted size of the protein). Why are all bands, including the one at 100kda (which is too high to be a product of the METTL3 gene), disappearing with sh METTL3 treatment?

We thank the reviewers for alerting us to this strange pattern. We re-analyzed all of our lysates from the METTL3 shRNA-treated cells by SDS-PAGE and immunoblot. We now show a blot that is representative of the results in general (Figure 2G). METTL3 migrates in these blots with the expected molecular weight as published by others.

4. Figure S2- in the original uncropped blots I'm unclear on what it is shown. Not all the blots have the same number of samples and only 2 lanes are circled in each. The expression of some proteins seems to be altered. This should be carefully quantified with adequate number of biological replicates, as this data might be important for nuclear retention result interpretation.

We repeated this experiment again and present new blots in Figure S2A with updated quantification in S2B.

5. Figure 3- If depleting YTHDC1/2 impacted the expression of ZFC3H1 and depleting ZFC3H1 impacted the expression of YTHDC1/2, these results should be carefully kept into consideration when interpreting any result.

We agree with the reviewer and we point this out in the text. However, due to space limitations, we do not comment further.

6. Since manipulation of ALKBH5 gave counter intuitive results that dispute the authors' hypothesis, it may be informative to test the effect of FTO.

This is an important point. We now write about this in the discussion.

November 20, 2024

RE: Life Science Alliance Manuscript #LSA-2024-03142-T

Dr. Alexander F Palazzo
University of Toronto
Biochemistry
1 King's College Circle MSB 5336
661 University Ave
Toronto, Ontario M5S 1A8

Dear Dr. Palazzo,

Thank you for submitting your revised manuscript entitled "N-6-methyladenosine (m6A) Promotes the Nuclear Retention of mRNAs with Intact 5' Splice Site Motifs". We would be happy to publish your paper in Life Science Alliance pending final revisions necessary to meet our formatting guidelines.

- please be sure that the authorship listing and order is correct
- please add a summary blurb and a category for your manuscript to our system
- please add the Twitter handle of your host institute/organization as well as your own or/and one of the authors in our system
- on page 7, you have a callout for Figure S1D-J, but there isn't a panel J in your Figure S1-please correct
- on page 9, you have a callout for Figure 3 H-I, but the panel I does not exist in Figure 3-please correct
- please add a figure callout for Figure 2 G-I to your main manuscript text
- a Data Availability statement should be added at the end of the Materials and Methods section to include the accession information for the miCLIP-seq dataset

A. FINAL FILES:

B. MANUSCRIPT ORGANIZATION AND FORMATTING:

Sincerely,

November 22, 2024

RE: Life Science Alliance Manuscript #LSA-2024-03142-TR

Dr. Alexander F Palazzo
University of Toronto
Biochemistry
MaRS Building, West Tower, Room 1533
661 University Ave
Toronto, ON M5G 1M1
Canada

Dear Dr. Palazzo,

Thank you for submitting your Research Article entitled "N-6-methyladenosine (m6A) Promotes the Nuclear Retention of mRNAs with Intact 5' Splice Site Motifs". It is a pleasure to let you know that your manuscript is now accepted for publication in Life Science Alliance. Congratulations on this interesting work.

*****IMPORTANT:** If you will be unreachable at any time, please provide us with the email address of an alternate author. Failure to respond to routine queries may lead to unavoidable delays in publication. *******

DISTRIBUTION OF MATERIALS:

Again, congratulations on a very nice paper. I hope you found the review process to be constructive and are pleased with how the manuscript was handled editorially. We look forward to future exciting submissions from your lab.

Sincerely,
